# Scaling Agent Learning via Experience Synthesis

**Zhaorun Chen**[1,3*]**, Zhuokai Zhao**[1]**, Kai Zhang**[1]**, Bo Liu**[2]**, Qi Qi**[1]**, Yifan Wu**[1]**,
Tarun Kalluri**[1]**, Sara Cao**[1]**, Yuanhao Xiong**[1]**, Haibo Tong**[3]**, Huaxiu Yao**[4]**, Hengduo Li**[1]**,
Jiacheng Zhu**[1]**, Xian Li**[2]**, Dawn Song**[5]**, Bo Li**[3]**, Jason Weston**[2†]**, Dat Huynh**[1†]

[1]Meta Superintelligence Labs, [2]FAIR at Meta, [3]University of Chicago, [4]UNC, [5]UC Berkeley

## Abstract

While reinforcement learning (RL) can empower autonomous agents by enabling self-improvement through interaction, its practical adoption remains challenging due to costly rollouts, limited task diversity, unreliable reward signals, and infrastructure complexity, all of which obstruct the collection of scalable experience data. To address these challenges, we introduce DREAMGYM, the first unified framework designed to synthesize diverse experiences with scalability in mind to enable effective online RL training for autonomous agents. Rather than relying on expensive real-environment rollouts, DREAMGYM distills environment dynamics into a reasoning-based experience model that derives consistent state transitions and feedback signals through step-by-step reasoning, enabling scalable agent rollout collection for RL. To improve the stability and quality of transitions, DREAMGYM leverages an experience replay buffer initialized with offline real-world data and continuously enriched with fresh interactions to actively support agent training. To improve knowledge acquisition, DREAMGYM adaptively generates new tasks that challenge the current agent policy, enabling more effective online curriculum learning. Experiments across diverse environments and agent backbones demonstrate that DREAMGYM substantially improves RL training, both in fully synthetic settings and in sim-to-real transfer scenarios. On non-RL-ready tasks like WebArena, DREAMGYM outperforms all baselines by over $30\%$. And in RL-ready but costly settings, it matches GRPO and PPO performance using only synthetic interactions. When transferring a policy trained purely on synthetic experiences to real-environment RL, DREAMGYM yields significant additional performance gains while requiring far fewer real-world interactions, providing a scalable warm-start strategy for general-purpose RL.

## 1 Introduction

Autonomous agents based on large language models (LLMs) are being widely adopted across a broad range of tasks given their comprehensive pre-trained semantic knowledge. These agents have already shown promise in applications such as web navigation (Zhou et al.), embodied control (Shridhar et al.), and multi-turn tool use (Yao et al., 2024). However, while these agents can leverage strong language priors to reason and plan, their performance in downstream interactive settings remains limited (Wang et al., 2024). As we step into the *era of experience* (Silver & Sutton, 2025), a promising direction for building more robust and adap-

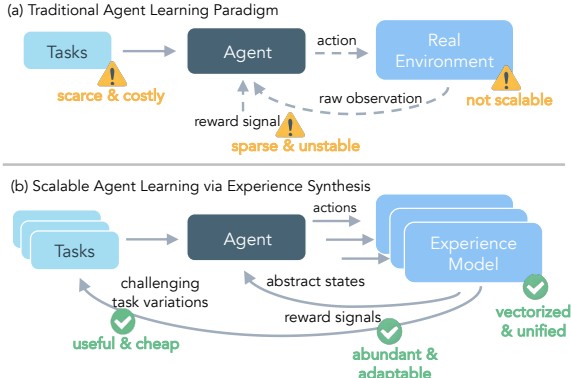

Figure 1: Compared to the traditional agent learning paradigm, DREAMGYM provides the first scalable and effective RL framework with unified infrastructure.

*Work done at Meta. †Joint last author. Correspondence to zhaorun@uchicago.edu, {jase, dathuynh}@meta.com.

tive language agents is reinforcement learning (RL), where agents improve by interacting with environments and bootstrapping from their own experiences (Schulman et al., 2017), as illustrated in figure 1 (a).

Despite its potential, training LLM agents via RL remains highly challenging in practice. The most fundamental barrier is the high cost and low sample efficiency of collecting large-scale, diverse, and informative online interaction data (Wei et al., 2025; Jiang et al., 2025). Real environments often involve long interaction sequences, high computational cost per step, and sparse reward feedback, making it prohibitively expensive to gather sufficient amount of data for modern RL algorithms (Patil et al.; Shao et al., 2024). Beyond computational cost, there is also a lack of diverse, scalable tasks, where most existing environments provide only a limited, static set of instructions, while RL training requires a broad range of tasks for effective exploration (Eysenbach et al., 2018). However, scaling task instructions is inherently difficult, as validating their feasibility often demands costly human expertise (Xue et al., 2025), leaving current environments insufficient for goal-conditioned RL. The third major barrier is the instability of reward signals. Many interactive settings, such as web pages and GUIs, are highly dynamic and lack consistent behaviors, resulting in noisy, sparse, or even false feedback that hinders stable learning (Deng et al., 2023). Safety concerns further compound these challenges, as certain actions are irreversible (e.g., deleting an item on a real website), and most environments lack reliable reset mechanisms (Zhou et al.). Finally, the infrastructure difficulty of constructing RL-ready environments also remain challenging. Existing systems are heterogeneous and often rely on heavyweight backends like Docker (Jimenez et al.) or virtual machines (Xie et al., 2024b), making large-batch rollout sampling engineering-intensive and costly. These limitations make building general-purpose and scalable systems for training agents with RL an open and pressing challenge.

To address these challenges, we propose **DREAMGYM**, a unified and scalable RL framework that synthesizes diverse experience data in an online manner to enable efficient and effective training of LLM agents. At the core of DREAMGYM lies a scalable *reasoning-based experience model* that abstracts environment dynamics into a discrete textual space. By interacting with the agent over multiple turns, it produces consistent transitions and feedback that reflect the consequences of the agent's actions through explicit reasoning. Unlike prior approaches that attempt to reproduce external systems (Chen et al., 2025; Assran et al., 2025), the design of the experience model is grounded in a key insight that *agent training does not require perfectly realistic environments, but rather interaction data that is sufficiently diverse, informative, and causally grounded to acquire knowledge* for the target task. Therefore, powered by strong reasoning, the experience model overcomes the key limitations outlined above and delivers useful experience data for RL training.

To ensure that synthetic experiences are diverse and informative, DREAMGYM equips the experience model with an *experience replay buffer*, from which it retrieves similar yet diverse trajectories to guide its current state prediction. This buffer is seeded with offline knowledge for essential context and is continuously enriched with trajectories generated on-the-fly, co-evolving the experience model with the agent to ensure the produced rollouts aligned with the agent's updated policy for stable training. In parallel, the experience model serves as a *task generator*, identifying valuable tasks with high reward entropy and producing progressively more challenging variations. This design yields an effective curriculum, where agents are consistently exposed to harder problems as their capability improves. By unifying interaction, memory, and adaptive online task generation, DREAMGYM addresses the persistent challenges that have limited RL for LLM agents training: prohibitive cost, scarcity of diverse tasks, unstable reward signals, and heavy infrastructure demands. It reframes training around an environment purpose-built for RL, enabling efficient synthetic training and effective sim-to-real transfer, improving generalization while minimizing reliance on costly real-world interactions.

Comprehensive experiments are conducted to evaluate DREAMGYM across diverse environments and LLM agent backbones. For use cases lacking RL training support (e.g., WebArena (Zhou et al.)), DREAMGYM provides the only viable approach for RL-based agent training, delivering over 30% improvement over all baselines and SOTA methods. In settings where RL is supported but costly, DREAMGYM achieves performance on par with GRPO (Shao et al., 2024) and PPO (Schulman et al., 2017), while training entirely within DREAMGYM without external interactions. Moreover, we introduce DREAMGYM-S2R (sim-to-real), which first trains agents in DREAMGYM using diverse, curriculum-driven synthetic experiences before transferring them to external environments. This approach yields over 40% performance improvement compared to training from scratch in real-world

environments while using less than 10% of the external data, providing a scalable warm-start strategy for general-purpose RL.

## 2 RELATED WORK

### 2.1 LLM AGENTS REINFORCEMENT LEARNING

RL offers a path to transform LLM agents from static generators into adaptive decision makers. Classical RL algorithms such as policy gradients and actor–critic methods (Williams, 1992; Schulman et al., 2017) have achieved strong results in robotics, games, and control (Silver et al., 2016). More recently, RL has been applied for post-training to improve their alignment (Bai et al., 2022) and general reasoning ability (Liu et al., 2025), as well as to enhance performance on math and coding problem-solving tasks (Shao et al., 2024).

However, extending RL from single-turn language modeling to interactive, multi-step agent tasks introduces substantial challenges (Zhang et al., 2025; Wang et al., 2025a). Tasks such as web navigation (Yao et al., 2022; Zhou et al.), operating-system control (Xie et al., 2024b), and multi-tool orchestration (Yao et al., 2024; Xie et al., 2024a) require long-horizon interaction yet provide sparse and delayed rewards (Qi et al., 2024), making policy improvement challenging and costly, and often result in training collapse. In addition, scaling RL for LLM agents is constrained by the lack of diverse, verifiable task suites, which typically require substantial human effort to design and annotate reward logic (Zhou et al., 2025). These challenges are further compounded by heterogeneous environment dynamics and engineering constraints in the real world. For example, open environments such as web pages are highly dynamic and produce noisy reward signals (Xue et al., 2025). Meanwhile, they often lack reliable reset mechanisms and introduce safety risks during large-scale exploration (Zhou et al.). Together, these challenges make real-environment RL impractical for general-purpose agents, motivating the need for a scalable solution that can supply adaptive, high-quality interaction data.

### 2.2 TRAINING AGENTS WITH SYNTHETIC DATA

Synthetic data has long been used to address the scarcity of expert demonstrations (Zhang et al., 2025). Early approaches relied on scripting oracle trajectories or generating them from stronger teacher models, training agents primarily through imitation learning (Yao et al., 2022; Pahuja et al., 2025; Deng et al., 2023). While effective for distillation, these static trajectories require substantial human labeling and lack diversity and adaptivity. To reduce manual effort, subsequent work (Xu et al.; Ou et al., 2024) leveraged indirect sources of expertise (e.g., online tutorials) to guide trajectory synthesis, while another line of work, including AgentSynth (Xie et al., 2025) and SCA (Zhou et al., 2025), focuses on synthesizing diverse tasks to expand the space of RL exploration. However, these methods still depend on data collection in real environments, which inherently suffer from the scalability issue and are subject to the limitations outlined earlier.

Later research shifted toward building synthetic environments, such as AlphaGo (Silver et al., 2016) and Dreamer (Hafner et al., 2020; Ha & Schmidhuber, 2018), which interact with agents to generate unlimited on-policy experiences. More recently, WebDreamer (Gu et al., 2024) and WebEvolver (Fang et al., 2025) constructed world models to produce environment feedback for agent planning and training. Closest to our setting, UI-Simulator (Wang et al., 2025b) also leverages LLMs as a step-wise simulator, but it requires substantial expert engineering to adapt to different environments and is limited to generating only trajectory variations for supervised fine-tuning of the policy. In contrast, DREAMGYM provides a complete toolkit for general-purpose RL, supporting diverse task generation and consistent rollout synthesis with rich reward signals, offering the first scalable solution for training generic LLM agents in arbitrary environments.

## 3 PRELIMINARIES

### 3.1 NOTATIONS

We formalize the agent learning problem as a Markov Decision Process (MDP) (Bellman, 1957), defined by the tuple $\mathcal{M} = (\mathcal{S}, \mathcal{A}, T, R, \gamma, \rho_0)$, where $\mathcal{S}$ denotes the state space and $\mathcal{A}$ denotes the action space. The transition function $T \colon \mathcal{S} \times \mathcal{A} \to \Delta(\mathcal{S})$ governs the environment dynamics, where $\Delta(\mathcal{S})$ denotes the probability simplex over $\mathcal{S}$. The reward function $R \colon \mathcal{S} \times \mathcal{A} \to \mathbb{R}$ provides feedback signals for the agent's actions. $\gamma \in [0, 1]$ is the discount factor, and $\rho_0 \in \Delta(\mathcal{S})$ specifies the initial state distribution that includes the task instruction $\tau_0$.

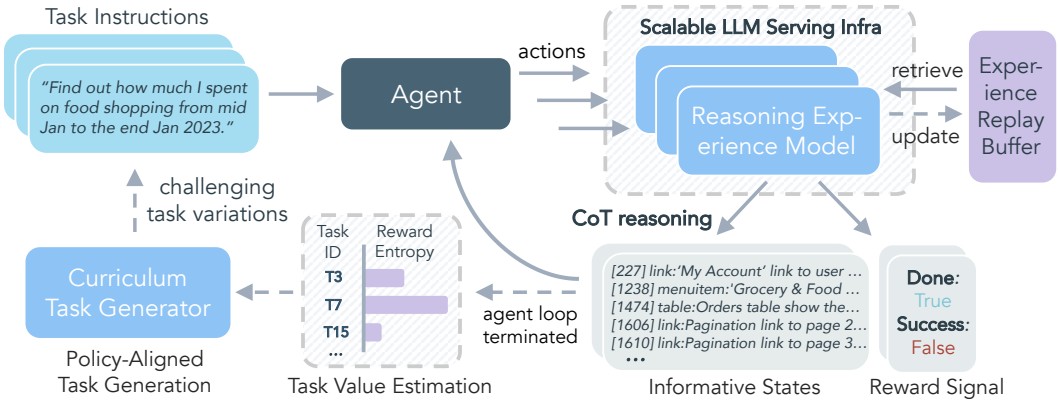

Figure 2: Overview of the proposed DREAMGYM agent training framework. Given a set of seed tasks, a reasoning-based experience model interacts with the agent to generate informative, diverse tasks and trajectories for RL training. At each step, the agent takes actions based on its current state and receives next states and reward signals derived by the experience model through CoT reasoning based on both interaction history and top-$k$ similar experiences from an active replay buffer. To expose the agent to increasingly informative scenarios, tasks with high reward entropy are proposed by the curriculum task generator for future training. With this unified design, DREAMGYM addresses both task and reward sparsity while enabling scalable RL with diverse and curriculum-driven environments.

In LLM agent environments, $\tau_0$ is usually a desired task specified by the user in natural language, and states $s \in \mathcal{S}$ encode the environment configuration visible to the agent, such as webpage content, tool outputs, or textual environment descriptions. Actions $a \in \mathcal{A}$ represent discrete operations, including clicking UI elements, invoking external tools, or generating textual responses. The agent maintains a policy $\pi_\theta \colon \mathcal{S} \to \Delta(\mathcal{A})$, parameterized by $\theta$, which maps states to distributions over actions.

### 3.2 AGENT LEARNING FROM EXPERIENCE

Given a set of online experiences where each experience $\epsilon = \{\tau_0 \mid s_0, a_0, \ldots\}$ consists of a task $\tau_0$ and state-action rollout $\{s_0, a_0, \ldots, s_t, a_t\}$, the goal of RL is to train an agent policy $\pi_\theta$ to maximize the expected cumulative reward, which typically optimized $\theta$ via policy gradient as follows:

$$\nabla J(\theta) = \mathbb{E}_{(s_t, a_t) \sim \pi_\theta} \left[ \nabla \log \pi_\theta(a_t \mid s_t) \cdot \hat{A}(s_t, a_t) \right], \tag{1}$$

where $\hat{A}(s_t, a_t)$ is the *advantage function*, estimating how favorable an action is compared to others.

**Proximal Policy Optimization (PPO).** PPO (Schulman et al., 2017) is a popular policy gradient method that improves stability by computing $\hat{A}$ with *Generalized Advantage Estimation (GAE)*:

$$\hat{A}_t^{\text{PPO}} = \sum_{l=0}^{K-1} (\gamma\lambda)^l \left[ r_{t+l} + \gamma V(s_{t+l+1}) - V(s_{t+l}) \right], \tag{2}$$

where $V(\cdot)$ is a value function approximated by a LLM, and $\lambda$ controls the bias-variance tradeoff.

**Group Relative Policy Optimization (GRPO).** GRPO (Shao et al., 2024) extends PPO by discarding the value function and normalizing advantages within each group of responses $\mathcal{G}$ sampled for the same task instruction. Instead of GAE, the group-relative advantage is defined as:

$$\hat{A}_t^{\text{GRPO}} = (r_t - \text{mean}_{i \in \mathcal{G}}(r_i)) / \text{std}_{i \in \mathcal{G}}(r_i) \tag{3}$$

where $r_t$ is the reward for output $o_t$, $\text{mean}_{i \in \mathcal{G}}(r_i)$ and $\text{std}_{i \in \mathcal{G}}(r_i)$ are mean and standard deviation of rewards from group $\mathcal{G}$. GRPO discards the value function and approximates the advantage using relative normalized rewards, making policy updates more scalable but potentially less sample-efficient. Notably, our proposed DREAMGYM is orthogonal to specific RL algorithms and focuses on scaling the synthesis of diverse, informative experiences, thereby amplifying the effectiveness of RL training.

## 4 SCALING AGENT LEARNING VIA EXPERIENCE SYNTHESIS

To synthesize diverse agent experiences for RL training, DREAMGYM is built around three key components: (1) a scalable *reasoning experience model* that encodes the meta-dynamics of the

target domain to efficiently generate informative trajectories; (2) an *experience replay buffer* that integrates offline environment knowledge with online synthetic transitions, co-evolving with the agent to stay aligned with its updated policy; (3) a *curriculum task generator* that produces progressively challenging variations of high-value tasks selected via a reward-entropy heuristic. We elaborate each component in the following sections.

## 4.1 BUILDING REASONING EXPERIENCE MODELS FOR AGENT LEARNING

For effective RL training, instead of relying on heterogeneous external environments that are costly to interact with and difficult to control, DREAMGYM adopts a more adaptive and controllable approach by building a LLM-based experience model that can efficiently interact with the agent over multiple turns to generate diverse experiences with consistent outcomes and rich feedback signals for learning.

Unlike prior data-hungry and costly approaches that build world models to replicate the real world in raw pixel spaces, we design an efficient reasoning experience model, denoted as $\mathcal{M}_{\text{exp}}$, that operates in an abstract, meta-representational textual space $\mathcal{S}$. The key insight is that synthesizing transitions in this abstract state space can reduce irrelevant dimensions and produce trajectories that are more informative and token-efficient than those derived from raw observations. For example, in a web shopping task, instead of processing raw HTML code, the experience model directly synthesizes clean element listings while discarding irrelevant structural artifacts such as headers and tags. This state-space design makes training the experience model highly sample-efficient, requiring only small pubic trajectory datasets in our experiments, while also enhancing the effectiveness of agent learning.

### 4.1.1 INFERENCE FOR EXPERIENCE ROLLOUT COLLECTION

Notably, we find that beyond the current state-action pair, three additional contexts are important for improving state quality: (1) *interaction history* $\{(s_i, a_i)\}_{t=0}^T$, which incorporates the past trajectory in the context window to help maintain state consistency across multiple turns; (2) *task instruction* $\tau$, which conditions the experience model on the current goal, enabling it to interpret actions w.r.t. task objectives and thereby predict both state transitions and rewards more accurately; (3) *past experiences*, which are top-$k$ demonstrations $\{d_j\}_{j=1}^k$ retrieved from the replay buffer based on semantic similarity with the state-action pair, i.e., $\{d_j\}_{j=1}^k = \text{Top}_k(\cos(\phi(s_t, a_t), \phi(s_i, a_i)))$, where $\phi(\cdot)$ denotes an arbitrary semantic encoder. Leveraging knowledge this way reduces hallucinations and improves factuality for knowledge-intensive state predictions. Therefore, given these inputs, the experience model predicts the next state $s_{t+1}$ and reward $r_{t+1}$ via chain-of-thought (CoT) (Wei et al., 2022):

$$(s_{t+1}, r_{t+1}) = \mathcal{M}_{\text{exp}}\Big(R_t \big| \{(s_i, a_i)\}_{i=0}^t, \{d_j\}_{j=1}^k, \tau\Big). \tag{4}$$

where $R_t$ is an explicit reasoning trace produced by the experience model that guides the state transition. With such reasoning, it predicts the most consistent and informative transition and feedback that reflects the consequence of the agent action. For example, if the action is invalid, it transitions to a failure state and assigns a zero reward to signal the error, and vice versa. In our experiments, following (Feng et al., 2025), we adopt an outcome-based reward scheme, assigning $r = 1$ only at the final step when the task is successfully completed and $r = 0$ in all other cases.

### 4.1.2 TRAINING EXPERIENCE MODELS TO REASON

Benefiting from the abstract state-space design, training the experience model is highly sample-efficient and requires only limited data from the real environment. In practice, abundant offline trajectory datasets from public benchmarks such as the WebArena Leaderboard are sufficient for training. Our experience model distills such offline knowledge and then serves as a bridge to interact with the agent online for RL training.

Concretely, given a trajectory dataset $\mathcal{D} = \{(s_t, a_t, s_{t+1}, r_{t+1})\}$, each transition is annotated with an explicit reasoning trace $R_t^*$ by LLM (prompt shown in Appendix D.1), which explains why the action $a_t$ taken in state $s_t$ consequently leads to the next state $s_{t+1}$ and reward $r_{t+1}$ given the available contexts. To distill this knowledge, we train $\mathcal{M}_{\text{exp}}$ via SFT with a joint objective over reasoning generation and next-state prediction:

$$\mathcal{L}_{\text{SFT}} = \mathbb{E}_{(s_t, a_t, s_{t+1}, R_t^*) \sim \mathcal{D}}\Big[-\log P_\theta(R_t^* \mid s_t, a_t, \mathcal{H}_t, \mathcal{D}_k) - \log P_\theta(s_{t+1} \mid s_t, a_t, R_t^*, \mathcal{H}_t, \mathcal{D}_k)\Big], \tag{5}$$

where $\mathcal{H}_t$ denotes the interaction history, $\mathcal{D}_k$ denotes the retrieved top-$k$ demonstrations, and $\theta$ denotes the parameters of $\mathcal{M}_{\text{exp}}$. This objective ensures that the model (i) learns to generate faithful reasoning traces that explain the causal effect of an action, and (ii) leverages these traces to predict consistent and informative next states. By doing so, the experience model not only imitates expert trajectories but also acquires the ability to generalize reasoning for novel rollouts during RL training.

## 4.2 CURRICULUM-BASED TASK GENERATION

Diverse, curriculum-aligned task instructions are important for RL agents to acquire knowledge (Zhou et al., 2025). However, scaling task collections is costly, as it requires significant human effort to verify the feasibility of each task in the target environment. DREAMGYM inherently alleviates this burden by adapting to arbitrary new tasks within the target domain through synthetic multi-turn transitions. Building on this capability, we propose *curriculum-based task generation*, where the same experience model actively generates new tasks as variations of a set of $m$ seed tasks:

$$\tau_t = \mathcal{M}_{\text{task}}(\{\tau_{t-1}^i\}_{i=1}^m), \tag{6}$$

where $\mathcal{M}_{\text{task}}$ shares parameters with $\mathcal{M}_{\text{exp}}$. Specifically, the seed tasks are chosen based on two criteria: (1) they are sufficiently challenging for the current agent policy, thereby maximizing information gain; (2) they are well-defined, such that unrealistic or malformed tasks can be discarded.

To satisfy both conditions, we introduce a *group-based reward entropy* as a criteria for selecting high-quality and challenging tasks. Formally, for a task $\tau$, we define its value

$$\mathcal{V}_\tau = \frac{1}{n} \sum_{i=1}^n \left(r^i - \bar{r}\right)^2, \quad \text{where } \bar{r} = \frac{1}{n} \sum_{i=1}^n r^i, \tag{7}$$

where $r^i$ are the outcome rewards from $n$ rollouts of task $\tau$ within the group $\mathcal{G}$. For GRPO, $\mathcal{G}$ can simply be the training group, while for PPO, tasks can be first clustered using a semantic embedder, and each cluster essentially forms a group $\mathcal{G}$ from which task variations can be generated. Notably, a non-zero variance in $\mathcal{G}$ indicates that the agent observes both successes and failures on the task, signaling that the task is **feasible yet challenging**. A task reaches maximum entropy when **successes and failures are evenly balanced** in $\mathcal{G}$, providing the greatest information gain for credit assignment. This observation is consistent with recent findings that LLMs learn most effectively from tasks of intermediate difficulty (Gao et al., 2025). Thus, by feeding such high-entropy tasks into $\mathcal{M}_{\text{task}}$, we generate progressively more challenging variations to enhance agent exploration and knowledge acquisition.

To stabilize training, we introduce a hyperparameter $\lambda$ that bounds the proportion of synthetic tasks sampled per iteration. This preserves sufficient coverage of the original task distribution while directing exploration toward the current policy's weakness regions for curriculum-based improvement.

## 4.3 LEARNING FROM SYNTHETIC EXPERIENCES

**Policy training in synthetic environments.** As shown in figure 2, DREAMGYM begins with a seed task set and generates multi-turn rollouts for each task by alternating between the agent policy, which selects actions from states, and the experience model, which predicts next states conditioned on the agent action, history, and task context (as in section 4.1.1). The collected rollouts are used with standard RL algorithms (as in section 3.2) to update the policy. After each iteration, the experience model augments the task set by generating variations of challenging tasks with high reward entropy (as in  section 4.2). This cycle of interaction, training, and curriculum expansion continues until convergence or a predefined training budget is reached. Furthermore, we provide an analytical lower bound of the policy improvement in real environments when training with purely synthetic experiences from DREAMGYM under trust-region assumptions, as detailed in Appendix C.1.

**Sim-to-real policy transfer.** We further extend DREAMGYM to a sim-to-real (S2R) setting, where the agent policy is first trained with synthetic experiences and then transferred to RL in real environments. Pretraining in synthetic environments expands exploration coverage across diverse tasks and allows the agent to acquire broad knowledge at low cost, providing a strong initialization that makes subsequent real-environment learning more sample-efficient (Da et al., 2025). To enable seamless transfer, we ensure consistency of the state space between synthetic and real environments by applying the same rule-based mapping function or a lightweight fine-tuned model (Lee et al.).

Table 1: Comparison of DREAMGYM with various agent training algorithms. We evaluate four groups: (i) *offline imitation learning algorithms*: SFT, DPO; (ii) *online RL algorithms in real-world environments*: GRPO, PPO; and (iii) DREAMGYM, where agents are trained via the same RL algorithms but with purely synthetic experiences; (iv) DREAMGYM-S2R, where agents are first trained with synthetic experiences and then transfer to RL in real environments. Real data indicates the number of individual transitions (a trajectory often has ∼10 steps). Best performance is bolded.

| Algorithm | Real Data | WebShop | | | ALFWorld | | | WebArena | | |
|---|---|---|---|---|---|---|---|---|---|---|
| | | L3.2-3B | L3.1-8B | Q2.5-7B | L3.2-3B | L3.1-8B | Q2.5-7B | L3.2-3B | L3.1-8B | Q2.5-7B |
| *Offline Imitation Learning* | | | | | | | | | | |
| SFT | 20K | 32.0 | 35.1 | 32.9 | 61.7 | 68.0 | 71.8 | 6.1 | 5.5 | 7.3 |
| DPO | 40K | 35.9 | 31.0 | 34.8 | 63.3 | 63.9 | 61.1 | 5.5 | 4.8 | 4.8 |
| *GRPO* | | | | | | | | | | |
| Traditional | 80K | 62.1 | 65.0 | 66.1 | **65.3** | 70.9 | 79.8 | 7.3 | 6.1 | 6.1 |
| DREAMGYM | 0 | 59.3 | 63.9 | 68.3 | 62.1 | 66.3 | 71.0 | 13.3 | 9.1 | **12.7** |
| DREAMGYM-S2R | 5K | **70.5** | **75.0** | **72.1** | 65.0 | **75.9** | **82.4** | **13.9** | **9.7** | 11.2 |
| *PPO* | | | | | | | | | | |
| Traditional | 80K | 59.9 | **64.2** | 68.1 | 47.0 | 72.9 | **81.1** | 6.7 | 4.8 | 7.3 |
| DREAMGYM | 0 | 60.5 | 58.1 | 65.0 | 40.5 | 70.8 | 72.7 | **14.5** | **10.9** | 10.0 |
| DREAMGYM-S2R | 5K | **66.0** | 63.9 | **73.7** | **49.1** | **73.3** | 79.9 | 13.3 | **10.9** | **13.9** |

# 5 EXPERIMENTS

## 5.1 EXPERIMENTAL SETUP

We evaluate DREAMGYM on a diverse suite of agentic benchmarks and LLM backbones of varying sizes and model families to assess its generalizability and effectiveness in supporting RL for generic agent tasks while reducing costly interactions.

**Evaluation environments.** We consider three challenging agent benchmarks that span diverse domains, complexities, and levels of RL readiness: (1) WebShop (Yao et al., 2022), which requires reasoning to refine search queries and accurately identify products to complete e-commerce tasks; (2) ALFWorld (Shridhar et al.), which involves multi-turn tool-based embodied control to navigate 3D environments; (3) WebArena-Lite (Zhou et al.), which offers realistic web interaction interfaces but is not RL-ready, as it inherently lacks scalable data collection and environment reset mechanisms while incurring high computational costs. This mixture of environments allows us to evaluate DREAMGYM both in settings where RL is feasible but computational expensive, and where RL training is not yet tractable.

**Agent backbones.** We instantiate agents from different model families and sizes: Llama-3.2-3B-Instruct, Llama-3.1-8B-Instruct (Grattafiori et al., 2024), and Qwen-2.5-7B-Instruct (Team, 2024).

**Baselines.** We consider two types of traditional training strategies for agents. (1) Offline imitation learning: supervised fine-tuning (SFT), direct preference optimization (DPO) (Rafailov et al., 2023); (2) Online RL in real environments (traditional): GRPO (Shao et al., 2024), PPO (Schulman et al., 2017).

**Implementation details.** All main results are reported with the experience model trained from Llama-3.1-8B-Instruct (see section 4.1). To demonstrate that DREAMGYM can be applied to different RL algorithms, we first evaluate both GRPO and PPO entirely within DREAMGYM, without any real interactions. We further evaluate a hybrid scenario, DREAMGYM-S2R, where synthetic training is followed by a small-scale RL phase that requires only a limited number of rollouts in the original environments, demonstrating the effectiveness of using DREAMGYM as a mid-training stage to improve both sample efficiency and the attainable performance after transfer. Detailed parameter settings for each scenario are provided in Appendix B.

## 5.2 MAIN RESULTS

**Non-RL-ready environment.** DREAMGYM demonstrates the most significant advantage in environments where large-scale RL infrastructure is not available such as WebArena (Zhou et al.). Unlike existing attempts that fail to make RL effective due to environment limitations, agents trained purely in DREAMGYM achieve success rates exceeding 30% across all backbones (Table 1), whereas zero-shot RL baselines suffer from limited exploration diversity and sparse reward signals in the original environments. These results demonstrate that DREAMGYM is not merely a surrogate for costly rollouts, but a mechanism that makes RL training feasible in domains that were previously intractable due to inherent task and engineering constraints.

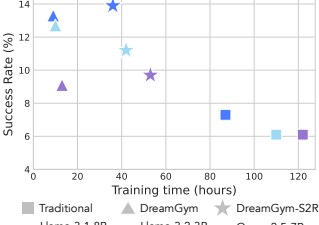 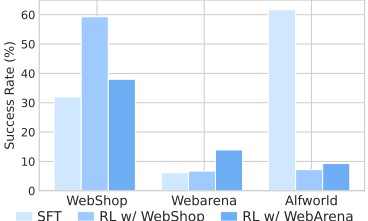 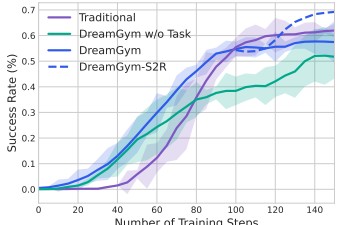

Figure 3: (1) *Left:* Comparing the agent performance (success rate %) on WebArena (Zhou et al.) w.r.t. total training time across different training strategies and backbones. (2) *Middle:* Evaluating the cross-domain transferability of the agent policy trained via DREAMGYM with seed tasks from a different environment. (3) *Right:* Comparing the agent performance on WebShop (Yao et al., 2022) w.r.t. number of training steps across different training strategies.

**RL-ready environments.** On WebShop (Yao et al., 2022) and ALFWorld (Shridhar et al.), DREAM-GYM-trained agents perform on par with GRPO and PPO agents trained on 80K real interactions, despite using only synthetic rollouts. The ability to match strong RL baselines in RL-ready environments without external interactions underscores that DREAMGYM produces transitions and rewards that are not only coherent and meaningful, but also sufficient for stable policy improvement. Furthermore, when a small-scale RL phase with an affordable number of real rollouts (5k) is applied to the policy mid-trained in the synthetic environment, DREAMGYM-S2R consistently outperforms both GRPO and PPO baselines trained from scratch in the real environments. This validates the hypothesis that synthetic training can serve as an efficient warm-start strategy that establishes a strong foundation for more sample-efficient RL in the real environment.

**Sample efficiency and training cost.** Training efficiency is further illustrated in figure 3 *Left*, where DREAMGYM achieves substantial performance gains on WebArena (Zhou et al.) while reducing training effort (including both rollouts sampling time and GPU hours) to roughly one-third or even one-fifth of RL baselines in the real environment. The efficiency gain arises from both the dense feedback offered by curriculum-based rollout synthesis and the lightweight abstract state transitions produced by unified experience models hosted by scalable LLM services, which substantially reduce sampling cost and avoid heterogeneous environment bottlenecks, suggesting that DREAMGYM is not only a practical solution for RL training in complex and expensive environments, but also as a scalable way to generate low-cost data for stable policy improvement.

**Generalization and learning transferability.** The results in figure 3 *Middle* highlight the strong generalization and cross-domain transferability of the policy trained in DREAMGYM. Notably, (1) when trained on WebShop (Yao et al., 2022), the policy generalizes to WebArena (Zhou et al.) and surpasses SFT models directly trained there, and (2) when trained on WebArena, it similarly transfers back to WebShop with superior performance than the SFT as well. This generalization suggests that DREAMGYM learns within an abstract meta-representation space, enabling the agent to learn domain-agnostic behavioral priors rather than memorizing task-specific patterns. However, (3) when the domain gap becomes too large, such as transferring from web-based environments (WebShop/WebArena) to ALFWorld (Shridhar et al.), the performance drops significantly, indicating the limits of current meta-representations.

# 6 PERFORMANCE ANALYSIS AND ABLATION STUDIES

## 6.1 TRAINING CURVE ANALYSIS

figure 3 *Right* compares training curves across Webshop (Yao et al., 2022) under different setups. Specifically, the success rate improves much more rapidly within the first 40 steps, showing that synthesized trajectories offer more informative gradients than sparse real rollouts. This further highlights the role of synthetic experiences in shaping a strong initialization that real rollouts cannot. Another observation is the reduced variance in learning dynamics. Baseline curves exhibit larger oscillations caused by sparse or unstable rewards, while DREAMGYM curves remain smoother across runs, which suggests that synthesized trajectories provide not only denser but also more consistent feedback, mitigating the training instabilities commonly reported in WebShop (Yao et al., 2022) and ALFWorld (Shridhar et al.).

## 6.2 ABLATION ON TASK GENERATOR

The curriculum-based task generator plays an important role in learning progress. As shown in figure 3 *Right*, removing this component causes agents to make some initial progress but then plateau more quickly in the WebShop (Yao et al., 2022) scenario. Similarly, Table 2 shows that removing the task generator leads to a 6.6% and 6.0% drop in success rate compared with the full DREAMGYM configuration in WebShop (Yao et al., 2022) and WebArena (Zhou et al.), respectively.

Table 2: Average success rates (%) on the different components of DREAMGYM.

| Method | WebShop | WebArena |
|---|---|---|
| DREAMGYM | 63.9 | 13.3 |
| w/o Exp. Replay | 59.2 | 9.7 |
| w/o Exp. Reasoning | 55.8 | 7.3 |
| w/o Task Generation | 57.3 | 7.3 |

These findings support our discussion in section 4.2: without adaptive task generation, the replay buffer may saturate with low-entropy, repetitive trajectories, which limits the diversity of experiences and stalls exploration. In contrast, the task generator continually produces progressively challenging, high-value tasks that push the agent beyond its current capability. This curriculum keeps the replay buffer informative and encourages exploration, yielding higher success rates and sample efficiency.

## 6.3 ABLATION ON EXPERIENCE MODEL

figure 4 demonstrates a detailed comparison of experiences generated by four variants of the experience models: a traditional real environment model, DREAMGYM, DREAMGYM without access to past trajectory history (w/o History), and DREAMGYM without reasoning (w/o Reasoning). We evaluate each variant along four criteria: consistency, diversity, informativeness, and hallucination, using GPT-4o (Hurst et al., 2024) as a judge. As detailed in Appendix D.4, the judge assigns discrete scores in $\{0, 1, 2\}$ for each criteria, where higher values indicate better performance. For the first three metrics, larger

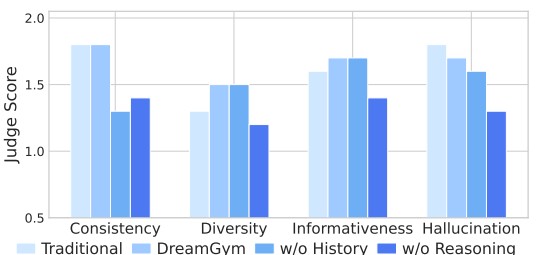

Figure 4: Evaluation of the experience model across key criteria using GPT-4o as the judge. We randomly sample 100 trajectories and prompt the model to assign discrete scores in $\{0, 1, 2\}$ across four criteria, as detailed in Appendix D.4.

scores mean more consistent, diverse, and informative; for hallucination, a score of 2 means no hallucination, while 0 indicates more factual errors.

The results highlight the role of each component. Removing trajectory history (w/o History) significantly reduces consistency: without awareness of prior turns, the model often drifts off-topic and breaks causal coherence in multi-step interactions. Removing reasoning (w/o Reasoning) mainly hurts informativeness of the states and increases hallucination: without reasoning capabilities, the generated experiences tend to become shallow and less factually grounded. Notably, removing experience reasoning also leads to a substantial drop in overall performance, as shown in Table 2. In contrast, the full DREAMGYM achieves the best or near-best performance across all metrics, confirming that history and reasoning provide complementary benefits. More specifically, history preserves temporal and causal structure, while reasoning enhances depth and factual reliability. This validates that the experience model must operate in a structured, reasoning-driven manner to maintain both diversity and fidelity of trajectories.

## 6.4 ABLATION ON EXPERIENCE MODEL BACKBONES AND OFFLINE TRAINING DATA

Notably, figure 5 investigates how the success rate of the experience model varies with both the amount of offline training data and the choice of model backbone, evaluated on (a) WebShop and (b) WebArena. We first observe that the experience model is highly data-efficient. Even with a very limited number of offline samples (2k-10k), it already reaches competitive performance. On WebShop, for example, the Llama-3.1-8B exceeds 50% success rate with only 10k samples, indicating that large-scale offline datasets are not strictly necessary for effective experience synthesis. Next, we find that smaller backbones remain viable. Although Llama-3.2-3B underperforms the 8B model, it improves steadily as more data becomes

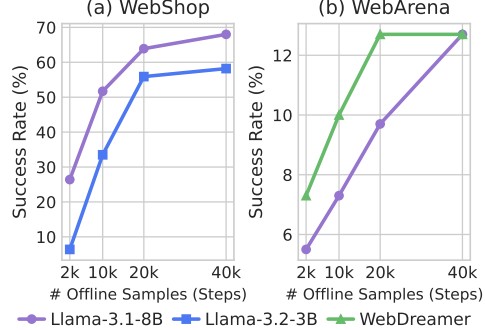

Figure 5: Evaluation of the experience model across different number of offline training data size (transition step) and backbone.

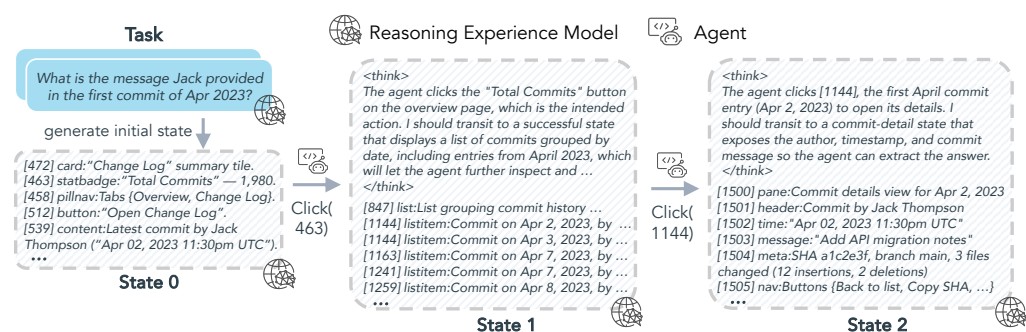

Figure 6: A case study of a trajectory sampled with DREAMGYM in WebArena. Starting from a synthetic instruction, the experience model reasons over the agent's action to produce future states.

available, reaching about 55% success on WebShop with 20k samples, which suggests that lightweight models can still serve as practical experience generators when computational resources are constrained. Finally, in the extreme low-data regime, pretrained world knowledge becomes particularly valuable. On WebArena, WebDreamer (Gu et al., 2024) (a fine-tuned web world model) achieves around 13% success, initially outperforming the Llama-3.1-8B model at smaller data scales but eventually converging as the number of offline samples increases. This suggests that while web-specific pretraining offers an early advantage, it is not a strict prerequisite, as larger-scale experience learning can enable pre-trained models to bridge the performance gap.

### 6.5 CASE STUDY OF SYNTHETIC EXPERIENCES FROM DREAMGYM

figure 6 illustrates how the reasoning experience model generates a synthetic task and progressively predicts states based on the agent's actions. Specifically, it predicts each state through explicit chain-of-thought reasoning that incorporates the agent's action, task instruction, and interaction history, producing next states that consistently ground the action and accurately reflect its consequences.

## 7 CONCLUSION

We introduced DREAMGYM, a framework that reduces the high cost of real-environment rollouts in RL for language agents by generating scalable, reasoning-driven synthetic experiences. DREAMGYM compresses environment dynamics into a reasoning-based experience model that produces state transitions and adaptive curricula, creating challenging yet solvable tasks tailored to the agent's evolving policy. Experiments across diverse environments and model backbones show consistent gains in both synthetic and sim-to-real settings, driven by the synergy of reasoning-based modeling, replay-buffer grounding, and curriculum generation. More broadly, our results suggest that the key bottleneck in RL for LLM agents lies in the quality and structure of interaction data. By treating environments as generators of structured, reasoning-rich experiences rather than mere simulators, DREAMGYM enables more scalable, sample-efficient, and generalizable RL for agents.

### ACKNOWLEDGEMENTS

We thank Sergey Levine, Xiaohan Fu, Canyu Chen for their valuable feedback on methodological conceptualizations, and Licheng Yu, Lizhu Zhang, Ravi Agrawal, Zhihan Liu, and Xiyao Wang for insightful discussions and project support.

This work is partially supported by the National Science Foundation under grant No. 1910100, No. 2046726, NSF AI Institute ACTION No. IIS-2229876, DARPA TIAMAT No. 80321, the National Aeronautics and Space Administration (NASA) under grant No. 80NSSC20M0229, ARL Grant W911NF-23-2-0137, Alfred P. Sloan Fellowship, the research grant from eBay, AI Safety Fund, Virtue AI, and Schmidt Science.

## USAGE OF LARGE LANGUAGE MODELS

The language in this paper was at times polished with the assistance of an LLM. The model was not used for research ideation, experimental design, or data analysis.

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

## A    APPENDIX

## B    DETAILED EXPERIMENT SETTINGS

In this section, we provide implementation details for each environment.

### B.1    WEBSHOP

WebShop (Yao et al., 2022) is a large-scale agent benchmark designed to study language grounding in interactive environments. It simulates a realistic e-commerce website with 1.18M real-world products and 12,087 crowd-sourced natural language instructions, where agents must search, customize, and purchase items. The environment poses challenges such as interpreting compositional product requirements, reformulating queries, handling noisy webpage text, and strategically exploring diverse page types.

**RL Baseline Setup.** We follow the standard setup and hyperparameter settings from Verl-Agent (Feng et al., 2025) and perform full-parameter fine-tuning for all three agent backbones in our experiments.

**DREAMGYM Settings.** To train the reasoning experience model, we construct a dataset by combining 1,600 human demonstration trajectories from the official WebShop repository with an additional 2,000 trajectories collected using an oracle agent and random exploration. We fix the test set to ensure evaluation stability and collect training trajectories only from the remaining tasks to avoid test-set contamination. Each transition is further augmented with a reasoning trace generated by a strong teacher LLM, resulting in the dataset used for fine-tuning.

**Computation Resources.** All experiments, including both baselines and ours, are conducted on 8 nodes with A100 GPUs and 4 nodes with H100 GPUs.

### B.2    ALFWORLD

ALFWorld (Shridhar et al.) is a text-and-embodied benchmark with hand-crafted task instructions designed for studying language grounding and cross-modal transfer. It pairs abstract text interactions from TextWorld with photo-realistic, physics-based execution in ALFRED/AI2-THOR, spanning six household task families (e.g., Pick & Place, Clean & Place, Heat/Cool & Place) with 3,553 training tasks and seen/unseen splits across 120 rooms. Agents issue high-level textual actions (goto, open, take, clean/heat/cool, put) that must be realized as low-level visuomotor controllers, facing challenges such as partial observability, object search and manipulation, mapping language to action preconditions and affordances, and bridging the gap between abstract plans and physical feasibility.

**RL Baseline Setup.** We adopt the standard setup and hyperparameter settings from Verl-Agent (Feng et al., 2025) and perform full-parameter fine-tuning for all three agent backbones.

**DREAMGYM Settings.** We follow the default `ALFWorld` split (Shridhar et al.) with the `TextWorld` setup (Côté et al., 2018) under the Verl-Agent framework. From the training split, we extract 3,200 expert demonstration trajectories paired with task instructions, and additionally sample 2,000 offline trajectories using both oracle and random policies. These datasets form the basis for training the reasoning experience model. Each transition is further augmented with a reasoning trace generated by a powerful LLM, which is used for fine-tuning.

**Computation Resources.** All experiments, including both baselines and ours, are conducted on 8 nodes with A100 GPUs and 4 nodes with H100 GPUs.

### B.3    WEBARENA

WebArena (Zhou et al.) is a self-hosted, realistic web environment for training and evaluating autonomous agents across fully functional sites such as e-commerce, social forums, collaborative software development (GitLab), and content management, which are augmented with tools (map, calculator, scratchpad) and knowledge bases (e.g., offline Wikipedia, manuals). It provides 812 long-horizon tasks expressed as high-level natural language intents and evaluates agents by functional correctness rather than matching action traces, supporting multi-tab browsing and a rich action space (click, type, navigate, tab operations).

**Training Set Split.** Since the full evaluation set in `WebArena` is large and contains many similar tasks, we follow prior work (Qi et al., 2024; Wei et al., 2025) and evaluate agents on `WebArena-Lite` (Liu et al., 2024), a more balanced subset of 165 high-quality, challenging tasks

selected from the original 812. The remaining 647 tasks, excluding those in the evaluation set, are used for training.

**RL Baseline Setup.** Since no reliable open-source RL infrastructure exists for WebArena (Qi et al., 2024), we follow the standard workflow in Verl-Agent (Feng et al., 2025) and implement a vanilla RL pipeline for WebArena. The baseline collects action trajectories via `browsergym` (Zhou et al.) while hosting the WebArena websites on AWS servers, supporting both PPO and GRPO. Despite extensive engineering effort, we are able to operate only four AWS servers, enabling at most four parallel interaction sessions, which in turn imposes a significant bottleneck on training throughput.

During RL sampling, we sequentially sweep through the task set and manually restart the servers and reset the environments once all tasks have been visited once, in order to avoid cross-task interference. Nevertheless, we observe that some trajectories fail to execute properly and that certain tasks are incorrectly judged by WebArena's original evaluation function, a known issue also reported in prior work (Qi et al., 2024; Chae et al., 2025). For fairness, we retain all collected trajectories and use them as-is for RL training.

**DREAMGYM Settings.** To obtain offline trajectories for training the reasoning experience model, we extract successful demonstrations from the highest-performing agents on the public WebArena leaderboard. Specifically, we use agents that incorporate accessibility tree information in their observations, including IBM CUGA (Marreed et al., 2025), ScribeAgent (Shen et al., 2024), Learn-by-Interact (Su et al., 2025), and AgentOccam (Yang et al., 2024). To mitigate distribution imbalance, we additionally collect trajectories generated by a high-performing agent and by a random policy. In total, we obtain 4,800 offline trajectories for training. Each transition is then augmented with a reasoning trace generated by a powerful LLM, forming the dataset for fine-tuning the reasoning experience model.

**Computation Resources.** All experiments, including both baselines and ours, are conducted on 8 nodes with A100 GPUs and 4 nodes with H100 GPUs.

## C  THEORETICAL ANALYSIS

In this section, we analyze how policies trained in the synthetic environments of DREAMGYM can provably improve performance in real environments. We show that, under mild assumptions, performance guarantees can be established by optimizing learning-centric signals of the experience model, such as reward accuracy and domain consistency, rather than strict fidelity metrics like state reconstruction error.

### C.1  PROVABLY POLICY IMPROVEMENT IN REAL ENVIRONMENTS TRAINED WITH SYNTHETIC EXPERIENCES

DREAMGYM trains LLM agents using a reasoning-based experience model $\mathcal{M}$exp, which interacts with the agent and induces a synthetic MDP $\widehat{\mathcal{M}}$. For brevity, we use $\widehat{\mathcal{M}}$ to denote any such synthetic environment, including $\mathcal{M}_{\exp}$, which is defined in the abstract textual state space, as stated in section 4.1. The learned policy is then evaluated in the real environment $\mathcal{M}$, projected into the same abstract space for comparison. We show that, under standard trust-region policy update assumptions, a policy optimized in $\widehat{\mathcal{M}}$ is guaranteed to also achieve policy improvement in the real environment $\mathcal{M}$.

**Theorem 1** (Policy Improvement $J$ in Real Environment via Synthetic Experiences). *Let the real MDP be $\mathcal{M} = (\mathcal{S}, \mathcal{A}, P, R, \gamma)$, the synthetic MDP induced by $\mathcal{M}_{\exp}$ be $\widehat{\mathcal{M}} = (\mathcal{S}, \mathcal{A}, \widehat{P}, \widehat{R}, \gamma)$, discount be $\gamma \in (0, 1)$, and let rewards be bounded $R, \widehat{R} \in [0, R_{\max}]$ with $V_{\max} := R_{\max}/(1 - \gamma)$. Assume one-step experience-model errors*

$$\varepsilon_R := \sup_{s,a} \big| R(s, a) - \widehat{R}(s, a) \big|, \qquad \varepsilon_P := \sup_{s,a} \mathrm{TV}\big( P(\cdot|s, a), \widehat{P}(\cdot|s, a) \big), \qquad (8)$$

*and a trust-region update $\pi \to \pi'$ obtained by optimizing in $\widehat{\mathcal{M}}$ with per-state KL radius $\sup_s D_{\mathrm{KL}}(\pi'(\cdot|s) \,\|\, \pi(\cdot|s)) \le \delta$, as enforced by the soft KL penalty in PPO and GRPO. Hence*

$$J_{\mathcal{M}}(\pi') - J_{\mathcal{M}}(\pi) \ge \underbrace{\frac{1}{1 - \gamma} \mathbb{E}_{s \sim d^{\pi}_{\widehat{\mathcal{M}}}, a \sim \pi'(\cdot|s)} \big[ A^{\pi}_{\widehat{\mathcal{M}}}(s, a) \big]}_{\text{synthetic surrogate gain in } \mathcal{M}_{\exp}} - \underbrace{\frac{4\gamma}{(1 - \gamma)^2} V_{\max} \delta}_{\text{trust-region penalty}} - \underbrace{2 \left( \frac{\varepsilon_R}{1 - \gamma} + \frac{2\gamma R_{\max}}{(1 - \gamma)^2} \varepsilon_P \right)}_{\text{experience model error}}$$
$$(9)$$

*In particular, if the synthetic surrogate gain exceeds the two penalties, then $J_{\mathcal{M}}(\pi') \ge J_{\mathcal{M}}(\pi)$.*

Specifically, (1) the *synthetic surrogate gain* denotes the agent's performance improvement when trained and evaluated within the synthetic environment provided by the experience model $\mathcal{M}_{\exp}$. (2) The *trust-region penalty* corresponds to the KL radius $\delta$ constraint, which is softly enforced by PPO or GRPO. (3) The *experience-model error* measures how well $\mathcal{M}_{\exp}$ preserves *learning-relevant signals* of the original environment for agent knowledge acquisition including two key components: (a) the **faithfulness of feedback** ($\varepsilon_R$), i.e., how accurately reward signals reflect real outcomes, and (b) the **domain consistency of state transitions** ($\varepsilon_P$), i.e., how well state space distributions align with the dynamics from the original environment.

Notably, these two error terms align with our design insights in section 4.1: **the synthetic environment need only provide domain-consistent transitions and correct, retrospective learning signals, without having to clone the original environment at the raw state level**. In practice, both $\varepsilon_R$ and $\varepsilon_P$ can be made very small even when $\mathcal{M}_{\exp}$ is trained with minimal trajectory data annotated with explicit reasoning traces.

*Proof of Theorem 1.* We first decompose the policy improvement in the real environment through the synthetic environment:

$$J_{\mathcal{M}}(\pi') - J_{\mathcal{M}}(\pi) = \big( J_{\widehat{\mathcal{M}}}(\pi') - J_{\widehat{\mathcal{M}}}(\pi) \big) + \big( J_{\mathcal{M}}(\pi') - J_{\widehat{\mathcal{M}}}(\pi') \big) - \big( J_{\mathcal{M}}(\pi) - J_{\widehat{\mathcal{M}}}(\pi) \big). \quad (10)$$

By Lemma 1, each of the two policy discrepancy terms $\big| J_{\mathcal{M}}(\cdot) - J_{\widehat{\mathcal{M}}}(\cdot) \big|$ is at most $\Delta_{\text{model}}$, hence

$$J_{\mathcal{M}}(\pi') - J_{\mathcal{M}}(\pi) \ge \big( J_{\widehat{\mathcal{M}}}(\pi') - J_{\widehat{\mathcal{M}}}(\pi) \big) - 2\Delta_{\text{model}}. \quad (11)$$

It remains to lower bound improvement inside the synthetic environment. Using the standard trust-region bound (Schulman et al., 2015), which is enforced in practice by PPO and GRPO via a per-state KL radius $\delta$, we have

$$J_{\widehat{\mathcal{M}}}(\pi') - J_{\widehat{\mathcal{M}}}(\pi) \ge \frac{1}{1 - \gamma} \mathbb{E}_{s \sim d^{\pi}_{\widehat{\mathcal{M}}}, a \sim \pi'(\cdot|s)} \big[ A^{\pi}_{\widehat{\mathcal{M}}}(s, a) \big] - \frac{4\gamma}{(1 - \gamma)^2} V_{\max} \delta. \quad (12)$$

Combining these two terms yields the inequality in Theorem 1, which completes the proof. $\square$

**Lemma 1** (Multi-turn experience synthesis error bound). *For any policy $\pi$, if*

$$\varepsilon_R = \sup_{s,a} |R(s,a) - \widehat{R}(s,a)|, \qquad \varepsilon_P = \sup_{s,a} \mathrm{TV}\big(P(\cdot|s,a), \widehat{P}(\cdot|s,a)\big), \tag{13}$$

*then*

$$\big|J_{\mathcal{M}}(\pi) - J_{\widehat{\mathcal{M}}}(\pi)\big| \leq \Delta_{\mathrm{model}} := \frac{\varepsilon_R}{1-\gamma} + \frac{2\gamma R_{\max}}{(1-\gamma)^2}\,\varepsilon_P. \tag{14}$$

*Proof.* We first compare the Bellman operators of the real and synthetic environments. For any bounded value function $V$,

$$(T_\pi V)(s) = \mathbb{E}_{a\sim\pi(\cdot|s)}\big[R(s,a) + \gamma\,\mathbb{E}_{s'\sim P(\cdot|s,a)}V(s')\big], \tag{15}$$

and let $\widehat{T}_\pi$ be the same expression with $(R, P)$ replaced by $(\widehat{R}, \widehat{P})$. Thus for any bounded value function $V$, the operator difference is bounded as

$$\|T_\pi V - \widehat{T}_\pi V\|_\infty \leq \sup_{s,a} |R(s,a) - \widehat{R}(s,a)| + \gamma \sup_{s,a} \Big|\mathbb{E}_{s'\sim P(\cdot|s,a)}V(s') - \mathbb{E}_{s'\sim\widehat{P}(\cdot|s,a)}V(s')\Big| \tag{16}$$

$$\leq \varepsilon_R + 2\gamma\|V\|_\infty\varepsilon_P, \tag{17}$$

which is derived by simply using the definitions of $\varepsilon_R, \varepsilon_P$ and the variational characterization of TV. Now apply this bound to $V = V_{\mathcal{M}}^\pi$ and add–subtract:

$$\|V_{\mathcal{M}}^\pi - V_{\widehat{\mathcal{M}}}^\pi\|_\infty = \|T_\pi V_{\mathcal{M}}^\pi - \widehat{T}_\pi V_{\widehat{\mathcal{M}}}^\pi\|_\infty \tag{18}$$

$$\leq \|T_\pi V_{\mathcal{M}}^\pi - \widehat{T}_\pi V_{\mathcal{M}}^\pi\|_\infty + \|\widehat{T}_\pi V_{\mathcal{M}}^\pi - \widehat{T}_\pi V_{\widehat{\mathcal{M}}}^\pi\|_\infty \tag{19}$$

$$\leq \varepsilon_R + 2\gamma V_{\max}\varepsilon_P + \gamma\|V_{\mathcal{M}}^\pi - V_{\widehat{\mathcal{M}}}^\pi\|_\infty. \tag{20}$$

By rearranging the contraction term into the left side, we have

$$(1-\gamma)\|V_{\mathcal{M}}^\pi - V_{\widehat{\mathcal{M}}}^\pi\|_\infty \leq \varepsilon_R + 2\gamma V_{\max}\varepsilon_P. \tag{21}$$

Hence

$$\|V_{\mathcal{M}}^\pi - V_{\widehat{\mathcal{M}}}^\pi\|_\infty \leq \tfrac{1}{1-\gamma}\big(\varepsilon_R + 2\gamma V_{\max}\varepsilon_P\big). \tag{22}$$

Finally, since $J_{\mathcal{E}}(\pi) = \mathbb{E}_{s_0\sim\mu}[V_{\mathcal{E}}^\pi(s_0)]$, we obtain

$$\big|J_{\mathcal{M}}(\pi) - J_{\widehat{\mathcal{M}}}(\pi)\big| = \Big|\mathbb{E}_{s_0\sim\mu}\big[V_{\mathcal{M}}^\pi(s_0) - V_{\widehat{\mathcal{M}}}^\pi(s_0)\big]\Big| \tag{23}$$

$$\leq \|V_{\mathcal{M}}^\pi - V_{\widehat{\mathcal{M}}}^\pi\|_\infty \tag{24}$$

$$\leq \frac{\varepsilon_R}{1-\gamma} + \frac{2\gamma R_{\max}}{(1-\gamma)^2}\varepsilon_P \tag{25}$$

$$=: \Delta_{\mathrm{model}}. \tag{26}$$

This indicates that the gap of agent performance between real and synthetic environments depends only on reward accuracy and domain consistency errors, rather than on strict fidelity metrics such as state reconstruction error, etc. $\square$

# D DREAMGYM PROMPTS

## D.1 WEBSHOP

---

**Experience model reasoning step annotation | `WebShop`**

| | |
|---|---|
| **System Prompt:** | You are an expert in web navigation and e-commerce environments, specializing in providing actionable guidance for state transitions of an experience model that simulates the environment dynamics. |
| **User Prompt:** | You are synthesizing environment state transition plans for training world models in webshopping tasks. You are provided with a task `instruction`, a `flag` indicating whether the trajectory is successful, and a `trajectory` $\{(s_i, a_i)\}_{i=1}^N$ of the environment state and the corresponding agent action at each step.
**Task Context**:
Task: {`instruction`} Success: {`flag`}
**Trajectory Steps:**
`" ".join(["Step: {i}, Environment State: {`$s_i$`}, Action: {`$a_i$`}"]`$_{i=1}^N$`)`
**Your Task**:

• **Task Tutorial**: A high-level guidance of how the environment should transit step-by-step to interact with the agent under the given task instruction. It should highlight the critical steps that the agent should perform in order for the environment to transit to the final successful state.
• **State Transition Plans**: For each step, first analyze whether the agent's action is likely to success or fail based on the task tutorial (e.g. the search query is too vague or too specific, or the agent clicks the wrong product), and then provide a concise reasoning trace describing how the environment should transition given the current state and action.

**CRITICAL**: You MUST generate exactly one transition plan for each environment step provided and your `state_transitions` array must contain exactly `len(env_step_ids)` entries, one for each `step_id`. For product listing pages, the state transition plan should mention some actionable details such as the number of products shown on this page, whether this page should contain the target product given the agent's action. Focus on actionable guidance for training the experience model. Keep responses concise and practical.

**Response Format**: json { "task_tutorial":
{"Overall Plan": "A one-sentence high-level guidance of how the environment should transit step-by-step to interact with the agent under the given task instruction.",
"Success Mode": "Describe the critical steps that the agent should perform to succeed in the task, where the environment should correspondingly transit to the successful state. Summarize in one sentence.",
"Failure Mode": "Describe the typical failure mode the agent should avoid, where the environment should correspondingly transit to the failed state once the agent performs the action. Summarize in one sentence." },
"state_transitions": [{
"step_id": 0,
"transition_plan": "Analyze whether the agent's action is good or bad based on the next state and overall task tutorial, and a corresponding plan for how environment should respond to this action."}

...
] } |

**Task variation dataset construction | `WebShop`**

| | |
|---|---|
| **System Prompt:** | You are an expert in e-commerce task design and AI training data curation. |
| **User prompt:** | You are an expert in e-commerce task design. I will give you an original web shopping `task instruction` and several `candidate variations` of this task. Your job is to select the most challenging yet feasible variation that would be good to train an AI agent to acquire the skills of shopping for the given `product`.

**Original Task**: {`task instruction`}
**Product Information**:

    1. Category: {`product_info['category']`}
    2. Product Name: {`product_info['name']`}
    3. Available Attributes:
{`','.join(product_info['attributes'])`}

**Candidate Variations**:{`candidates variations`}
**Criteria for selection**:

    • **Challenging but Feasible**: The task should be more specific or complex than the original, but still achievable, so as to strengthen the agent's capabilities for shopping for the given product.
    • **High Quality**: The instruction should be clear, grammatically correct, and realistic.
    • **Meaningful Variation**: The changes should make the task meaningfully different (not just trivial changes).
    • **Realistic**: The combination of attributes, options, and price should make sense for the product category.

**Please respond with**:

    1. The number of your selected variation 1- (`len({candidate variations})`).
    2. A brief explanation (1-2 sentences) of why this variation is the most challenging and high-quality.

**Format your response as**:
SELECTION: [number]
REASONING: [explanation] |

**Agent Prompt Template | `WebShop`**

You are an expert autonomous agent operating in the WebShop e-commerce environment. Your task is to:
`{task_description}`.

Prior to this step, you have already taken {`step_count`} step(s). Below are the most recent {`history_length`} observations and the corresponding actions you took:
`{action_history}`
You are now at step {`current_step`} and your current observation is:
`{current_observation}`.
Your admissible actions of the current situation are:
`[{available_actions}]`.

Now it's your turn to take one action for the current step. You should first reason step-by-step about the current situation, then think carefully which admissible action best advances the shopping goal. This reasoning process MUST be enclosed within `<think>` `</think>` tags.
Once you've finished your reasoning, you should choose an admissible action for current step and present it within `<action>` `</action>` tags.

## D.2 ALFWORLD

---

**Task variation dataset construction | `ALFWorld`**

| | |
|---|---|
| **System Prompt:** | You are an expert in embodied task design and AI training data curation for interactive embodied environments. |
| **User prompt:** | You are an expert in embodied task design. I will give you a feasible `task instruction` for an embodied agent and several `candidate variations` of this task. Your job is to select the most challenging yet feasible variation that would be good to train an AI agent to acquire generalizable embodied reasoning skills. |

**Original Task**: {`task instruction`}
**Environment Context**:

1. Room Type: {`env_info['room']`}
2. Objects Present: {`','.join(env_info['objects'])`}
3. Containers/Surfaces:
{`','.join(env_info['locations'])`}

**Candidate Variations**: {`candidate variations`}
**Criteria for selection**:

- **Challenging but Feasible**: The variation should add complexity (e.g., more objects, constraints, or multi-step actions) without being impossible.
- **High Quality**: Clear, grammatical, and realistic in the ALFWorld context.
- **Meaningful Variation**: Should involve non-trivial differences in *action type*, *target object*, or *target location*.
- **Realistic**: The variation must be consistent with ALFWorld's embodied environment dynamics (e.g., no placing a fridge on a lamp).

**Please respond with**:

1. The number of your selected variation 1-(`len({candidate variations})`).
2. A brief explanation (1-2 sentences) of why this variation is the most challenging and high-quality.

**Format your response as**:
SELECTION: [number]
REASONING: [explanation]

---

**Agent Prompt Template | `ALFWorld`**

You are an expert agent operating in the ALFRED Embodied Environment.
Your task is to:
{`task_description`}

Prior to this step, you have already taken {`step_count`} step(s). Below are the most recent {`history_length`} observations and the corresponding actions you took:
{`action_history`}.
You are now at step {`current_step`} and your current observation is:
{`current_observation`}.
Your admissible actions of the current situation are:
[{`admissible_actions`}].

Now it's your turn to take an action. You should first reason step-by-step about the current situation. This reasoning process MUST be enclosed within `<think>` `</think>` tags. Once you've finished your reasoning, you should choose an admissible action for current step and present it within `<action>` `</action>` tags.

## D.3 WEBARENA

---

**Task variation dataset construction | `WebArena`**

| | |
|---|---|
| **System Prompt:** | You are an expert in designing realistic, diverse, and challenging web interaction tasks for AI agents. |
| **User prompt:** | I will provide you with several seed `WebArena task instructions`. Your job is to generate new task variations from each seed. The variations should keep the **same** general action type (e.g., search, filter, upvote, navigate, purchase, delete) but **differ** in target, constraints, or context, making them realistic, challenging, and meaningfully different. |

**Seed Instructions**: {`list of seed instructions`}

**Requirements for variations**:

- **Action Consistency**: Preserve the same type of action as the seed task.
- **Meaningful Differences**: Change the entities, filters, domains, time ranges, or constraints so the new task is distinct but natural.
- **Challenging but Feasible**: The variation should slightly increase reasoning or constraint complexity, but remain solvable.
- **High Quality**: Grammatically correct, clear, and realistic web tasks.

**Please respond with**:

For each seed instruction, generate [K] new task variations. Format your response as:
SEED: [original instruction]
VARIATIONS:
    1. [variation 1]
    2. [variation 2]
...

**Example**:

SEED: List products from living room furniture category by descending price.
VARIATIONS:
    1. List products from bedroom furniture category by ascending price.
    2. Show me the most expensive three dining tables available online.
    3. Find discounted sofas under $500 in the living room furniture category.

---

---

**High-quality task variation selection | `WebArena`**

| | |
|---|---|
| **System Prompt:** | You are an expert in interactive web task design and AI training data curation. |
| **User prompt:** | You are an expert in web environment task design. I will give you an original WebArena `task instruction` and several `candidate variations` of this task. Your job is to select the most challenging yet feasible variation that would be good to train an AI agent to acquire generalizable skills in web interaction.
**Original Task**: {`task instruction`}
**Candidate Variations**: {`candidate variations`}
**Criteria for selection**:

    • **Challenging but Feasible**: The variation should require slightly more reasoning, precision, or constraints than the original, but still be solvable by a web agent.
    • **High Quality**: Clear, grammatical, and realistic within the web environment.
    • **Meaningful Variation**: Keep the same *action type* (e.g., search, navigate, sort, submit, upvote, purchase) as the original, but change the context, target, or condition.
    • **Realistic**: The task should reflect plausible web interactions a user might request.

**Please respond with**:

    1. The number of your selected variation 1-(`len({candidate variations})`).
    2. A brief explanation (1-2 sentences) of why this variation is the most challenging and high-quality.

**Format your response as**:
SELECTION: [number]
REASONING: [explanation] |

---

**AX-tree state mapping prompt | `WebArena`**

| | |
|---|---|
| **System Prompt:** | You are an agent tasked with extracting and refine a subset of the webpage's observations based on the content of the page and user instructions. Perform the following tasks based on the provided [Information source], including user instructions, interaction history, and the AXTree observation at the current time step. First, provide high-level reasoning for the next action by analyzing the provided information. Second, extract relevant webpage elements based on your high-level reasoning. |
| **User prompt:** | [General instructions]
You are currently on the `{domain_info}` website. Your task is to generate a **Reasoning** and a **Refined observation** based on the provided inputs.
First, review the **User instruction** and **History of interactions** and, then, generate the **Reasoning**. Analyze the progress made so far, and provide a rationale for the next steps needed to efficiently accomplish the user instruction on the `{domain_info}` website.
Second, refine the **Webpage observation at the current time step** into a **Refined observation**. Extract a subset of the webpage observation (e.g., chart, table, menu items) that contains necessary information for completing the user instruction, and explain the extracted elements. Ensure that the information on the elements (e.g., numeric element ID) is correctly included.
Please follow the format in the [Reasoning & Refinement example] carefully.

[Information source]
**User instruction**: `{user instruction}`
**History of interactions**:`{interaction history}`
**Webpage observation at the current time step**:`{AXTree observation}`

[Reasoning & Refinement example]
**Abstract example**
Here is an abstract version of the answer, describing the content of each tag. Make sure you follow this structure and format strictly, but replace the content with your own answer:
`<reasoning>`
Think step by step. Based on the **User instruction**, **History of interaction**, and **AXTree observation at the current time step**:

    • Provide a high-level description of the **AXTree observation at the current time step.**
    • Based on the **User instruction** and **History of interaction** track your progress and provide your
       reasoning on the next action needed to accomplish the **User instruction**

Ensure that: Structure your reasoning concisely and follow the following format strictly:
`<content_description>` High-level description of current page state (max 2 sentences)`</content_description>` `<agent_progress>` What has been accomplished so far (max 1 sentence)`</agent_progress>` `<next_action_analysis>` What should happen next and why (max 1 sentence)`</next_action_analysis>`
`</reasoning>`
`<extraction>`
Based on your reasoning, identify the elements (e.g., buttons, text fields, static text, table row, chart) to focus on. Then, explain the semantics and functionalities of each extracted element. Ensure that: You do not alter the structure of the AXTree observation. You extract the element ID (id in [ ]) accurately without any errors. When extracting chart or table, you must extract the entire chart or table to avoid any confusion or loss of information. Unless necessary, try not to extract url or non-semantic identifiers which is not informative for the agent actions. All the elements you extract should be actionable and discard irrelevant elements. Please follow the following format and do not provide any other text besides the element list.
[ELEMENT_ID] TYPE:DESCRIPTION
[ELEMENT_ID] TYPE:DESCRIPTION
...
[ELEMENT_ID] TYPE:DESCRIPTION
(Extract 3-10 most relevant actionable elements only)
`</extraction>` |

---

**Agent Prompt Template | `WebArena`**

| | |
|---|---|
| **System Prompt:** | You are an agent trying to solve a web task based on the content of the page anda user instructions. You can interact with the page and explore. Each time you submit an action it will be sent to the browser and you will receive a new page. |
| **User prompt:** | **Instructions** |

Review the current state of the page and all other information to find the best possible next action to accomplish your goal. Your answer will be interpreted and executed by a program, make sure to follow the formatting instructions.

**User instruction**: {user instruction}
**History of interactions**:{interaction history}
**Refined observation of current step**: Reasoning {plan}
**Focused AXTree observation**: {rep_observation}
**Action space**: 13 different types of actions are available.

- noop(wait_ms: float = 1000)
    1. Description: Do nothing, and optionally wait for the given time (in milliseconds).
    2. Examples: noop(),noop(500)
- ...

To save space, please refer to D.3.1 for the full list of actions.

**Remark:** Only a single action can be provided at once. Example:`fill('a12', 'example with "quotes"')` Multiple actions are meant to be executed sequentially without any feedback from the page. Don't execute multiple actions at once if you need feedback from the page.

**Abstract Example**
Here is an abstract version of the answer with description of the content of each tag. Make sure you follow this structure, but replace the content with your answer:
`<think>`
Think step by step. If you need to make calculations such as coordinates, write them here. Describe the effect that your previous action had on the current content of the page.
`</think>`
`<action>`
One single action to be executed. You can only use one action at a time.
`</action>`

**Concrete Example**
Here is a concrete example of how to format your answer. Make sure to follow the template with proper tags:
`<think>`
My memory says that I filled the first name and last name, but I can't see any content in the form. I need to explore different ways to fill the form. Perhaps the form is not visible yet or some fields are disabled. I need to replan. `</think>`
`<action>`
`fill('a12', 'example with "quotes"')`
`</action>`

### D.3.1 ACTION SPACE OF WEBARENA

- **noop** `(wait_ms:  float = 1000)`
    1. Description: Do nothing, and optionally wait for the given time (in milliseconds).
    2. Examples: `noop(); noop(500)`
- **send_msg_to_user** `(text:  str)`
    1. Description: Send a message to the user. You should send a short answer as a message and do not ask questions through message.
    2. Examples: `send_msg_to_user('the city was built in 1751.');` `send_msg_to_user('Yes'); send_msg_to_user('No');` `send_msg_to_user('31112'); send_msg_to_user('Yoshua Bengio')`
- **scroll** `(delta_x:  float, delta_y:  float)`
    1. Description: Scroll horizontally and vertically. Amounts in pixels, positive for right or down scrolling, negative for left or up scrolling. Dispatches a wheel event.
    2. Examples: `scroll(0, 200); scroll(-50.2, -100.5)`
- **fill** `(bid:  str, value:  str)`
    1. Description: Fill out a form field. It focuses the element and triggers an input event with the entered text. It works for `<input>`, `<textarea>` and `[contenteditable]` elements.
    2. Examples: `fill('237', 'example value'); fill('45', 'multi-line example'); fill('a12', 'example with "quotes"')`
- **select_option** `(bid:  str, options:  str | list[str])`
    1. Description: Select one or multiple options in a `<select>` element. You can specify option value or label to select. Multiple options can be selected.
    2. Examples: `select_option('48', 'blue'); select_option('48', ['red', 'green', 'blue'])`
- **click** `(bid:  str, button:  Literal['left', 'middle', 'right'] = 'left', modifiers:  list[typing.Literal['Alt', 'Control', 'Meta', 'Shift']] = [])`
    1. Description: Click an element.
    2. Examples: `click('51'); click('b22', button='right'); click('48', button='middle', modifiers=['Shift'])`
- **dblclick** `(bid:  str, button:  Literal['left', 'middle', 'right'] = 'left', modifiers:  list[typing.Literal['Alt', 'Control', 'Meta', 'Shift']] = [])`
    1. Description: Double click an element.
    2. Examples: `dblclick('12'); dblclick('ca42', button='right'); dblclick('178', button='middle', modifiers=['Shift'])`
- **hover** `(bid:  str)`
    1. Description: Hover over an element.
    2. Examples: `hover('b8')`
- **press** `(bid:  str, key_comb:  str)`
    1. Description: Focus the matching element and press a combination of keys. It accepts the logical key names that are emitted in the `keyboardEvent.key` property of the keyboard events: `Backquote, Minus, Equal, Backslash, Backspace, Tab, Delete, Escape, ArrowDown, End, Enter, Home, Insert, PageDown, PageUp, ArrowRight, ArrowUp, F1 - F12, Digit0 - Digit9, KeyA - KeyZ, etc.` You can alternatively specify a single character you'd like to produce such as `"a"` or `"#"`. Following modification shortcuts are also supported: `Shift, Control, Alt, Meta`.
    2. Examples: `press('88', 'Backspace'); press('a26', 'Control+a');` `press('a61', 'Meta+Shift+t')`
- **focus** `(bid:  str)`
    1. Description: Focus the matching element.
    2. Examples: `focus('b455')`
- **clear** `(bid:  str)`
    1. Description: Clear the input field.
    2. Examples:`clear('996')`
- **drag_and_drop** `(from_bid:  str, to_bid:  str)`
    1. Description: Perform a drag & drop. Hover the element that will be dragged. Press left mouse button. Move mouse to the element that will receive the drop. Release left mouse button.
    2. Examples: `drag_and_drop('56', '498')`
- **upload_file** `(bid:  str, file:  str | list[str])`
    1. Description: Click an element and wait for a "filechooser" event, then select one or multiple input files for upload. Relative file paths are resolved relative to the current working directory. An empty list clears the selected files.
    2. Examples: `upload_file('572', 'my_receipt.pdf'); upload_file('63', ['/home/bob/Documents/image.jpg', '/home/bob/Documents/file.zip'])`

D.4   EXPERIENCE MODEL JUDGE

---

**Judge Prompt for Evaluating Experience Models**

You are an expert judge for scoring the quality of a predicted state transition in a WebShop environment simulator.
**You are given**:
- Current state (before the action)
- The agent action
- Predicted next state (after the action)

**Your task**:
1) Evaluate the predicted next state on **four rubrics**, each scored **0**, **1**, or **2**.
2) Provide brief step-by-step reasoning for **each rubric**.
3) Output a **valid JSON** object with the rubric scores and the total (sum of the four rubrics). Do not include extra fields.

**General rules**:
- Base your judgment **only** on the provided inputs; do not assume hidden context.
- Use integers only ($0/1/2$) for rubric scores.
- If an action is invalid or should not change the page, correct behavior may include a no-op with an explicit failure/empty-result signal.
- Be concise but specific in your reasoning (1–3 sentences per rubric). —
**# # # Rubrics ($0/1/2$) with anchors:**
1) **Causal State Consistency** | *Question:* Is the predicted next state both logically consistent with the prior state **and** causally grounded in the agent's action semantics (e.g., click → detail page, pagination → new results, search → updated listings, back → prior view)?

   - **2**: Coherent and action-appropriate; all expected updates appear with no contradictions.
   - **1**: Mostly consistent, but has minor logical or semantic gaps.
   - **0**: Inconsistent or not causally linked to the action.

2) **Diversity & State Variation** | *Question:* Is there a meaningful, non-degenerate change from the prior state (when change is expected)?

   - **2**: Substantive, coherent differences (new results, updated filters, changed details).
   - **1**: Minimal or superficial change.
   - **0**: No meaningful change, or incoherent jump.

3) **Informativeness** | *Question*: Is the predicted state rich, relevant, and internally coherent (e.g., listings with meaningful attributes; filters aligned with content)?

   - **2**: Detailed, relevant, and coherent information.
   - **1**: Some useful details, but sparse or partially incoherent.
   - **0**: Uninformative, irrelevant, or incoherent.

4) **Hallucination & Failure Feedback** | *Question*: When the action is invalid or yields no results, does the state reflect an appropriate failure/empty-result signal instead of hallucinating success?

   - **2**: Correctly signals failure or success as appropriate, no hallucination.
   - **1**: Partial/ambiguous handling of failure.
   - **0**: Hallucinates success or ignores failure.

—

**### Step-by-step Evaluation (use this structure):**
1. **Causal State Consistency:** `<your reasoning>` **Score:** 0/1/2
2. **Diversity & State Variation:** `<your reasoning>` **Score:** 0/1/2
3. **Informativeness**: `<your reasoning>` **Score:** 0/1/2
4. **Hallucination & Failure Feedback**: `<your reasoning>` **Score:** 0/1/2

—
**# # # Final JSON Output:**
Output a single valid JSON object. Replace angle brackets with integers only.

```
{"rubric_scores":  {
"causal_consistency":  <0|1|2>, "diversity":  <0|1|2>,
"informativeness":  <0|1|2>, "hallucination":  <0|1|2> }}
```

