# OpenReview forum: "Scaling Agent Learning via Experience Synthesis"
_ICLR.cc/2026/Conference — ICLR 2026 Poster_

### Official Review · Reviewer_PzPS · 2025-10-28

**Soundness:** 2
**Presentation:** 2
**Contribution:** 3
**Rating:** 6
**Confidence:** 4

**Summary:**

This paper introduces DreamGym, a framework designed to synthesize diverse experiences with scalability in mind to enable effective online RL training for autonomous agents. This method embeds transition knowledge into the a experience model and leverage it to generate new challenging task. To verify their method, the author conduct some experiments on several agentic tasks like Webshop.

**Strengths:**

1. The idea of experience model is novel.
2. The paper is well written.
3. The idea of embedding transition dynamic is good.

**Weaknesses:**

1. The baselines are too simple. There are many intrinsic rewards designed in reinforcement learning to encourage exploration, we recommend you include at least one.
2. Moreover, depending success label to select task group is not feasible for most situations. There can be a lot of circumstances in RL, The agent performs correctly in the first half of the task, but it is still making mistakes in the second half, in which case the task will not selected because the variation is zero.
3. No task generation example is provided. I expect to see some instances of the selected task groups and the generated new task from task model.

**Questions:**

See weaknesses. If none of the concerns is addressed，I will consider to downgrade the score.

---

> ### Author Response · Authors · 2025-11-23
>
> Dear Reviewer PzPS,
>
> We sincerely thank the reviewer for appreciating the novelty and effective design of our proposed method. We have carefully followed your suggestions by conducting additional experiments against the RL baselines and concurrent approaches, providing further clarification of the task-generation mechanism, and adding more illustrative examples of the generated tasks. These revisions have been incorporated into the paper and are summarized in the responses below. We hope they adequately address your concerns.
>
>
>
>
> > W1: The baselines are too simple. There are many intrinsic rewards designed in reinforcement learning to encourage exploration, we recommend you include at least one.
>
>
>
> **R1:** We sincerely appreciate the reviewer for raising this important point, and we fully agree that additional comparisons with representative RL algorithms and peer approaches help better contextualize the advantages and unique contribution of our method. Following the reviewer’s suggestion, we have expanded our evaluation to include three major groups of baselines:
> (1) representative generic RL algorithms beyond DPO, PPO, and GRPO already included in the paper, such as DAPO and Dr. GRPO, which introduce additional techniques for more efficient and stable training;
> (2) domain-specific RL algorithms on the WebArena environment that incorporate substantial engineering effort and intrinsic reward designs tailored to the web domain; and
> (3) concurrent synthetic-training approaches that share a similar motivation with DreamGym. Below, we elaborate on each set of results.
>
>
>
> ### **Comparison with Generic RL Algorithms**
>
> We compare DreamGym against a broader set of representative generic RL baselines, including DAPO [1] and Dr. GRPO [2], which are developed based on GRPO to improve its efficiency and training stability, representing the most widely used RL paradigms for LLM agent post-training. As shown in Table-r.1, both algorithms struggle in long-horizon, real-environment settings due to infrastructure bottlenecks and low sample efficiency, which is consistent to our findings in Table 1 from the main paper. In contrast, DreamGym achieves substantially higher success rates on WebArena than both DAPO and Dr. GRPO. Specifically, this improvement is achieved without any real online interactions, as DreamGym conducts the majority of training in its synthetic environment. On AlfWorld, DreamGym similarly maintains strong performance, highlighting its ability to scale to multi-step embodied environments.
>
> These results demonstrate that DreamGym can serve as a more scalable and effective RL training paradigm even when compared to strong general-purpose RL algorithms.
>
>
>
> ### **Comparison with Domain-Specific RL Frameworks**
>
> We also evaluate DreamGym against domain-engineered RL frameworks, i.e., WebAgent-R1 [3], WebRL [4], and DigiRL [5], which incorporate substantial task-specific engineering (e.g., custom intrinsic rewards, handcrafted wrappers, and reset mechanisms) designed specifically for WebArena. However, these frameworks are not transferable to other environments such as AlfWorld.
>
> From the results shown in Table-r. 1, we can draw the following conclusions:
>
> - DreamGym performs comparably to these domain-engineered SOTA systems while using no real online trajectories, whereas WebAgent-R1 and WebRL each require thousands of online interactions with the real environment to achieve similar performance, despite their intrinsic reward designs and substantial engineering effort, making them expensive and difficult to scale.
> - DreamGym-S2R, after a modest amount of continued training in real environments, outperforms WebAgent-R1 in success rate, despite requiring far fewer real interactions and incurring dramatically lower engineering complexity.
> - Since domain-specific methods are tightly coupled to WebArena infrastructure, they cannot run on AlfWorld, while DreamGym generalizes seamlessly.
>
> These findings highlight that DreamGym achieves SOTA-level performance without heavy domain specialization, offering far broader applicability across environments.

---

> > ### Author Response · Authors · 2025-11-23
> >
> > **Table-r. 1** Comparison of DreamGym against RL algorithms including (1) generic RL baselines (e.g. DAPO and Dr. GRPO) and (2) domain-engineered RL frameworks (e.g. WebAgent-R1, WebRL, DigiRL) on WebArena and AlfWorld. Metrics include averaged success rate (%), estimated time cost (hours), and the total number of real trajectories for training.
> > | Method       | WebArena     |          |                      | AlfWorld     |          |                      |
> > | ---------------- | ---------------- | -------- | -------------------- | ---------------- | -------- | -------------------- |
> > |                  | Success Rate | Time | #Real Traj | Success Rate | Time | #Real Traj |
> > | DAPO         |      10.0            |  120        |      1200 online                |           66.1       |    50      |          4800 online            |
> > | Dr. GRPO     |      12.7            |   120       |     1200 online                 |          70.8        |    50       |     4800 online                  |
> > | WebAgent-R1  |      44.8            |  70        |    2400 online + 9460 offline                  | NA           | NA   | NA               |
> > | WebRL        |     42.4             |   80       |  4000 online + 12200 offline                    | NA           | NA   | NA               |
> > | DigiRL        |     30.3             |  80        |  4000 online + 12200 offline                     | NA           | NA   | NA               |
> > | DreamGym     |      45.0            |   15       |    4800 offline                  |        70.8          |   30        |    5200 offline                    |
> > | DreamGym-S2R |      48.3            |   55       |     800 online + 4800 offline                 |    73.3               |  42       |  1600 online +  5200 offline                  |
> >
> >
> > ### **Comparison with Concurrent Synthetic-Training Approaches**
> >
> > To further highlight DreamGym's design advantages, we also compare it against two concurrent frameworks that train agents in synthetic environments, Simia-RL [6] and UI-Simulator [7], both released after our initial submission. Results are shown in Table-r. 2 and Appendix C.2. The key observations are:
> >
> > - Compared to UI-Simulator, which directly synthesizes raw DOM trees using a strong reasoning LLM, DreamGym achieves much higher success rates and lower time costs by synthesizing transitions in an explicit, concise, token-efficient abstract state space. This significantly reduces inference overhead and improves the coherence of environment dynamics.
> > - Compared to Simia-RL, which relies on closed-source models such as OpenAI o4-mini to generate step-wise state predictions, DreamGym reduces monetary cost by over 30× while achieving higher success rates. This advantage arises because DreamGym’s experience model is fine-tuned on domain-specific offline trajectories, enabling it to produce more consistent, in-distribution transitions that directly support effective policy improvement.
> >
> >
> >
> >
> > **Table-r. 2** Comparison of DreamGym against concurrent synthetic training frameworks on the WebArena environment. Metrics include averaged success rate (%), estimated time cost (hours), number of real-world trajectories used for training, and the monetary expense of environment service (USD) such as AWS server hosts and API key costs.
> > | Method | Success Rate |Time | #Real Traj| Expense|
> > |--------|---------|----------|-----------|-----------|
> > | Traditional       |15.7 |120 | 1200 online | 272 |
> > | Simia-RL    | 37.0 | 17 |4800 offline  |460.8 |
> > | UI-Simulator |19.7 | 22 | 4800 offline | 72 |
> > | DreamGym    |45.0 | 15 | 4800 offline | 14.4 |
> >
> >
> > Therefore, by modeling the environment in a learning-oriented, token-efficient meta-representation space, DreamGym is more cost-effective by design and produces experience data that directly supports policy improvement without incurring unnecessary fidelity-related overhead.
> >
> >
> >
> > We sincerely hope that the expanded comparisons across generic RL baselines, domain-specific RL frameworks, and concurrent synthetic-training approaches could address the reviewer’s concerns and help clarify the novelty, generality, and unique contribution of our work.

---

> > > ### Author Response · Authors · 2025-11-23
> > >
> > > > W2: Moreover, depending success label to select task group is not feasible for most situations. There can be a lot of circumstances in RL, The agent performs correctly in the first half of the task, but it is still making mistakes in the second half, in which case the task will not selected because the variation is zero.
> > >
> > > **R2:** We thank the reviewer for raising this insightful question. We fully agree that, in many RL scenarios, an agent may perform well in the earlier part of a task but still make mistakes later on, leading to an overall failure outcome. In such cases, all sampled rollouts for that task would share the same failure label, resulting in zero variation under our outcome-based grouping.
> > >
> > > We would like to emphasize, however, that the purpose of our task-selection mechanism is to identify tasks that are *meaningfully challenging yet feasible* for the agent’s **current** learning stage, i.e., tasks of intermediate difficulty. The variation of success labels serves as a practical and readily available proxy for identifying such tasks in environments that only provide **outcome-based rewards**, as is the case in our experimental setting. Under this reward structure, the final success flag is the only cheap and observable signal we can rely on to estimate whether a task is on the border of the agent’s current capability range.
> > >
> > > As a matter of fact, what the reviewer suggests is aligned with the design principles of our approach. For example, tasks where the agent consistently fails (even if early steps appear correct) are treated as *too difficult* for the current policy. These tasks will naturally have zero success variation and are therefore not selected at this stage. As the agent improves, tasks that were previously too difficult will likely reach a point where at least some sampled rollouts succeed. At that moment, these tasks will automatically gain non-zero variation and become eligible for generating more challenging task variations, forming a natural curriculum without requiring additional supervision.
> > >
> > > In summary, although outcome-based variation is a coarse signal, our group-based reward entropy serves as a reliable and practical proxy in this setting to select challenging tasks for the agent. It effectively allows DreamGym to construct an adaptive curriculum that naturally progresses from easier to harder tasks using a simple, low-cost mechanism. That said, we fully agree with the reviewer that it would be beneficial to incorporate tasks where the agent succeeds only in part of the trajectory, which can be better supported in environments with procedural or step-wise rewards that explicitly label intermediate correctness.

---

> > > > ### Author Response · Authors · 2025-11-23
> > > >
> > > > > W3: No task generation example is provided. I expect to see some instances of the selected task groups and the generated new task from task model.
> > > >
> > > >
> > > > **R3:** We thank the reviewer for the interest in seeing concrete examples of generated tasks. While we already provide some examples in Figure 6 of the paper that illustrates the full synthesis pipeline that starts from a generated task and show the following agent–environment interactions, we completely agree that additional explicit examples of task generation will help clarify the mechanism. Below we provide representative samples from each environment by randomly sampling one original task and showing several new variations generated by the experience model.
> > > >
> > > > ---
> > > >
> > > > ### Example 1: WebShop
> > > >
> > > > We randomly sampled a challenging task from WebShop that requires the agent to locate a tareget product by navigating through the webpages, parsing attributes, and applying multi-constraint filtering:
> > > >
> > > > **Original task:** *Find me home office furniture sets for dining room, living room with color: blue, and item shape: runner, and size: 3 ft 7 in x 5 ft 2 in, and price lower than 70.00 dollars.*
> > > >
> > > > Using the experience model, DreamGym generated the following variations:
> > > >
> > > > + **Product variation**: *Find me area rugs for dining room and living room with color: blue, item shape: runner, size: 3 ft 7 in x 5 ft 2 in, and price lower than 70.00 dollars.*
> > > >
> > > > + **Product attribute variation:** *Find me home office furniture sets for dining room and living room with color: navy blue, item shape: rectangle, size: 2 ft 6 in x 8 ft 6 in, and price lower than 70.00 dollars.*
> > > >
> > > > + **Product price variation:** *Find me home office furniture sets for dining room and living room with color: blue, item shape: runner, size: 3 ft 7 in x 5 ft 2 in, and price between 40.00 and 60.00 dollars.*
> > > >
> > > > These variations preserve the structural semantics of the original task while introducing controlled perturbations, enabling the agent to learn tighter generalization over product attributes and constraints.
> > > >
> > > > ---
> > > >
> > > > ### Example 2: ALFWorld
> > > >
> > > > Unlike WebShop, ALFWorld tasks follow shorter, imperative natural-language instructions. We sampled the following original task:
> > > >
> > > > **Original task:**  *Heat some apple and put it on the countertop.*
> > > >
> > > > Our experience model produced the following variations:
> > > >
> > > > + **Variation in target object**: *Heat some tomato and put it on the countertop.*
> > > > + **Variation in target destination**: *Heat some apple and place it inside the cabinet.*
> > > > + **Variation in target action**: *Slice the apple and place the pieces on the countertop.*
> > > >
> > > >
> > > > These newly generated tasks (none of which appear in the original training set) expand the agent’s behavioral competence by varying objects, target destinations, and required actions, while staying grounded in ALFWorld’s affordance constraints and environment semantics.
> > > >
> > > > ---
> > > >
> > > > ### Example 3: WebArena
> > > >
> > > > Tasks in WebArena often involve complex reasoning, cross-page navigation, and actionable decision-making. For example, we sampled the following task instruction:
> > > >
> > > > **Original task:**  *Draft an email to the shop owner via their “Contact Us’’ function to request a coupon since my refund was supposed to be issued as store credit.*
> > > >
> > > > Then the experience model generated the following variations:
> > > >
> > > > + **Contextual rephrase:** *Write a polite email to the shop owner via their “Contact Us’’ function explaining that I shop regularly at this store and would prefer receiving the reimbursement as loyalty credit rather than a refund.*
> > > >
> > > > + **Request variation:**  *Contact the store via email to request a return address so that I can send the product back for return processing.*
> > > >
> > > > + **Action variation:** *Send a message through the store’s embedded support chat asking whether they could provide a 15% discount code for future purchases as compensation for the refund issue.*
> > > >
> > > >
> > > > These variations preserve the essential intent to communicate with a vendor through a web interface, while altering the situational framing, request content, and action channel. This enables the agent to generalize over multiple ways of expressing similar intents and strengthens robustness in real interaction scenarios.
> > > >
> > > >
> > > > We hope these additional examples provide clearer insight into how DreamGym generates new challenging tasks beyond the original tasks to construct an adaptive training curriculum and strengthen agent policy over time.

---

> > > > > ### Author Response · Authors · 2025-11-23
> > > > >
> > > > > We sincerely thank the reviewer for their thoughtful feedback and appreciation of our work. We hope that our responses have satisfactorily addressed your concerns, and we would also be glad to provide further clarification on any remaining questions. Thank you again for your time and efforts!
> > > > >
> > > > > Best regards,
> > > > >
> > > > > Submission #19123 Authors
> > > > >
> > > > > [1] Yu, Q., Zhang, Z., Zhu, R., Yuan, Y., Zuo, X., Yue, Y., ... & Wang, M. (2025). Dapo: An open-source llm reinforcement learning system at scale. arXiv preprint arXiv:2503.14476.
> > > > >
> > > > > [2] Liu, Z., Chen, C., Li, W., Qi, P., Pang, T., Du, C., ... & Lin, M. (2025). Understanding r1-zero-like training: A critical perspective. arXiv preprint arXiv:2503.20783.
> > > > >
> > > > > [3] Wei, Z., Yao, W., Liu, Y., Zhang, W., Lu, Q., Qiu, L., ... & Li, L. (2025). Webagent-r1: Training web agents via end-to-end multi-turn reinforcement learning. arXiv preprint arXiv:2505.16421.
> > > > >
> > > > > [4] Qi, Z., Liu, X., Iong, I. L., Lai, H., Sun, X., Zhao, W., ... & Dong, Y. (2024). Webrl: Training llm web agents via self-evolving online curriculum reinforcement learning. arXiv preprint arXiv:2411.02337.
> > > > >
> > > > > [5] Bai, H., Zhou, Y., Pan, J., Cemri, M., Suhr, A., Levine, S., & Kumar, A. (2024). Digirl: Training in-the-wild device-control agents with autonomous reinforcement learning. Advances in Neural Information Processing Systems, 37, 12461-12495.
> > > > >
> > > > > [6] Li, Y., Inan, H. A., Yue, X., Chen, W. N., Wutschitz, L., Kulkarni, J., ... & Rajmohan, S. (2025). Simulating Environments with Reasoning Models for Agent Training. arXiv preprint arXiv:2511.01824.
> > > > >
> > > > > [7] Wang, Y., Yin, D., Cui, Y., Zheng, R., Li, Z., Lin, Z., ... & Chang, K. W. (2025). LLMs as Scalable, General-Purpose Simulators For Evolving Digital Agent Training. arXiv preprint arXiv:2510.14969.

---

### Official Review · Reviewer_SnpT · 2025-10-30

**Soundness:** 3
**Presentation:** 3
**Contribution:** 3
**Rating:** 6
**Confidence:** 3

**Summary:**

This paper introduces DREAMGYM, a scalable framework for training LLM-based agents without heavy environment interaction. Instead of collecting real rollouts, the framework employs a reasoning-based experience model that generates synthetic trajectories through reasoning, simulating next states and rewards. It further incorporates curriculum-based task generation, selecting tasks with high reward entropy to maximize information gain and progressively increase difficulty. Experiments on WebShop, ALFWorld, and WebArena show that DREAMGYM achieves comparable or superior performance to PPO and GRPO while using less than 10% of real environment interactions.

**Strengths:**

- Targets a practical yet challenging problem in LLM agent RL — reducing training cost, wall-clock time, and instability arising from expensive environment interactions.
- The paper is well-structured and clearly written: the problem is precisely defined, and each subproblem is addressed through three clearly delineated components — experience model inference, experience model training, and curriculum-based task generation. It makes the overall paper easy to follow, even for non-expert readers.
- The experimental design is extensive and well-organized, effectively addressing questions that naturally arise during reading.

**Weaknesses:**

- Potential error accumulation in self-training loop: Since the experience model continuously generates synthetic rollouts and refines itself using those same transitions, any early bias or inaccuracy in its learned dynamics may compound over iterations. While this may not be severe in relatively simple domains (as Figure 4 suggests small gaps between real and synthetic transitions), it could become problematic in more complex or long-horizon environments where early modeling errors propagate and amplify.

- Unclear buffer update strategy: Are all synthetic transitions stored, or only feasible, high-quality ones are retained? Without a clear filtering or prioritization mechanism, low-quality or hallucinated transitions could accumulate in the buffer, potentially degrading both policy learning and experience model stability over time.

**Questions:**

See weaknesses above.

---

> ### Author Response · Authors · 2025-11-23
>
> Dear Reviewer SnpT,
>
>
> Thank you for your appreciation of the value and novelty of our work. We have carefully followed your suggestions, added further discussion of the potential error-accumulation issue in the self-training process, provided additional clarification on the buffer update strategy, and included new ablation experiments to demonstrate the practical effectiveness of our approach. We sincerely hope these additions address your concerns.
>
>
>
>
> > W1: Potential error accumulation in self-training loop: Since the experience model continuously generates synthetic rollouts and refines itself using those same transitions, any early bias or inaccuracy in its learned dynamics may compound over iterations. While this may not be severe in relatively simple domains (as Figure 4 suggests small gaps between real and synthetic transitions), it could become problematic in more complex or long-horizon environments where early modeling errors propagate and amplify.
>
> R1: We deeply appreciate the reviewer for raising this important and insightful question. Indeed, we totally agree with the reviewer that if not handled carefully, a self-training loop can suffer from error accumulation, where early modeling inaccuracies compound over iterations. However, DreamGym explicitly addresses this through a buffer update strategy designed to ensure both correctness and challenging level of the retrieved synthetic rollouts used for informing the in-context next-state prediction. Below, we clarify this mechanism in detail.
>
> To ensure that synthesized experiences co-evolve with the agent’s improving policy, DreamGym updates the replay buffer after each training epoch. The key idea is to refresh the buffer with experiences that are both **meaningful** (i.e., states where the agent is currently weak and tends to fail) and **feasible** (i.e., transitions that are grounded and non-hallucinatory). Concretely, based on the insight that a single state may lead to many potential terminal outcomes, given all the trajectories generated by the agent in each iteration, we apply a top-down search procedure to identify high-entropy, informative state–action pairs. Specifically:
> (i) similar to our entropy-based task-selection strategy, we keep only those transitions whose intermediate states have **led to at least one successful terminal outcome** among all subsequent rollouts, and
> (ii) we discard states that lead exclusively to failure across all rollouts, as they are more likely to reflect infeasible or ill-defined transitions.
>
>
> This criterion naturally selects feasible, diverse, and high-entropy states: a state that leads to at least one successful outcome is demonstrably valid, yet it may also lead to failure depending on the agent’s subsequent choices, therefore capturing both difficulty and correctness without risking error compounding.
>
> In addition, we further employ two specific techniques to reduce error accumulation and ensure robustness throughout both the buffer update stage and the retrieval exploitation stage.
>
> _(1) Update stage controlled via synthetic data ratio $\beta$_
> We explicitly cap the proportion of synthetic transitions allowed in the replay buffer by a hyperparameter $\beta$, defined relative to the number of real trajectories. Specifically, after each training epoch, if the number of synthetic transitions is below $\beta$, we fill the remaining slots using the filtered synthetic transitions from the current iteration. And if the buffer is already full, we discard the oldest synthetic experiences and refill them with new ones selected via the feasibility criteria above.
>
> This mechanism ensures that the buffer maintains a controlled and healthy ratio of synthetic data, preventing the synthetic distribution from becoming overly dominant and enabling effective quality control during self-training on model-generated transitions.
>
>
> _(2) Exploitation stage controlled via reweighted sampling_
> When retrieving state–action demonstrations for next-state prediction during inference, we apply a reweighted sampling strategy over the top-k most similar transitions. First, we begin by sampling according to similarity-based probabilities. Then, each time a synthetic example is selected, we downweight the probability of selecting additional synthetic examples in subsequent sampling rounds. In practice, this strategy can effectively prevent the agent from relying overly on synthetic data and enforces a balanced mix of real and synthetic demonstrations when informing the next-state prediction. Together, these two mechanisms significantly reduce the likelihood of error accumulation during the self-training process.

---

> ### Author Response · Authors · 2025-11-23
>
> To further address the reviewer’s concern, we conducted an ablation study evaluating the effect of the synthetic-data ratio $\beta$ of the replay buffer on both end-to-end agent performance and synthetic state quality. Consistent to Section 6.3, state quality is assessed using GPT-4o as a judge, including key criteria such as hallucination, consistency, diversity, and informativeness, where a higher score indicates the state is more consistent, diverse, and factual w.r.t. the real counterpart. We present the evaluation results on WebArena in Table-r. 1 below. When $\beta$ = 0.3, the agent achieves the highest success rate while maintaining the same state-quality score as $\beta$ = 0 (only real data, where no error accumulation during self-training is guaranteed), suggesting that introducing up to 30% synthetic data does not cause additional hallucination or observable error accumulation. Even at $\beta$ = 0.5, the state quality score remains high and does not significantly affect the agent's end-to-end learning effectiveness.
>
> These findings demonstrate that our buffer update and retrieval mechanisms effectively maintain the quality of the synthetic transitions in the replay buffer, while still allowing them to be challenging and co-evolve with the agent’s improving policy.
>
> ---
>
> **Table-r. 1** Ablation on the ratio of synthetic data $\beta$ in the replay buffer. We report end-to-end success rate on WebArena and the average state-quality score evaluated by GPT-4o as judge.
> | Method | Success Rate | State Judge Score |
> |--------|--------------|-------------------|
> | β = 0   | 42.0 | 1.6 |
> | β = 0.1 | 43.7 | 1.6 |
> | β = 0.3 | 45.0 | 1.6 |
> | β = 0.5 | 42.9 | 1.5 |
>
> ---
>
> In addition, to further showcase the effectiveness of the active self-training design of DreamGym, we also compare DreamGym with two concurrent synthetic-training frameworks: Simia-RL [1] and UI-Simulator [2] (both released after our initial submission). Results are shown in Table-r.2 below and in Appendix C.2 (Table 6) of the revised paper. Both of these approaches follow the high-level design principles of DreamGym in that they maintain a replay buffer to guide state prediction; however, their replay buffers contain only real offine data, which limits the diversity and informativeness of retrieved examples and constrains the agent’s learning efficiency. In contrast, DreamGym actively refreshes its replay buffer with feasible and challenging synthetic experiences, enabling it to continually co-evolve with the agent’s policy and provide up-to-date supervision during training. As shown in the results, agents trained in DreamGym outperform both Simia-RL and UI-Simulator in success rate, enabled by our co-evolving environment setup that continually adapts to the agent’s improving policy. At the same time, DreamGym incurs substantially lower time and monetary costs by synthesizing transitions in a concise, explicit, and token-efficient abstract state space, which greatly reduces environment-inference overhead and produces coherent, policy-relevant transitions. By contrast, UI-Simulator suffers from high latency due to generating raw DOM observations, and Simia-RL incurs more than 30× higher monetary cost because of its dependence on closed-source, high-capacity models.
>
> ---
>
> **Table-r. 2** Comparison of DreamGym against concurrent synthetic training frameworks on the WebArena environment. Metrics include average success rate (%), estimated time cost (hours), number of real-world trajectories used for training, and the monetary expense of environment service (USD) such as AWS server hosts and API key costs.
> | Method | Success Rate |Time | #Real Traj| Expense|
> |--------|---------|----------|-----------|-----------|
> | Traditional       |15.7 |120 | 1200 online | 272 |
> | Simia-RL    | 37.0 | 17 |4800 offline  |460.8 |
> | UI-Simulator |19.7 | 22 | 4800 offline | 72 |
> | DreamGym    |45.0 | 15 | 4800 offline | 14.4 |
>
>
> We hope this expanded explanation, together with the supporting ablations, clearly addresses the reviewer’s concern regarding the replay buffer update mechanism and how it mitigates potential error accumulation in the self-training loop. We sincerely appreciate the opportunity to clarify this important component of our framework.

---

> > ### Author Response · Authors · 2025-11-23
> >
> > > W2: Unclear buffer update strategy: Are all synthetic transitions stored, or only feasible, high-quality ones are retained? Without a clear filtering or prioritization mechanism, low-quality or hallucinated transitions could accumulate in the buffer, potentially degrading both policy learning and experience model stability over time.
> >
> > **R2:** We thank the reviewer for raising this concern. We have already provided a detailed explanation of the replay buffer update strategy and how it prevents low-quality or hallucinated transitions from accumulating in our response to **W1**, where we describe the full filtering, feasibility checks, and prioritization mechanisms to ensure the high-quality and utility of the actively updated experience data in the buffer.
> >
> > In summary, DreamGym does not store all synthetic transitions. Instead, it applies a lightweight filtering process to ensure that only feasible, grounded, and policy-active synthetic rollouts are updated into the buffer. Specifically, the strategy cover the following steps: (1) Feasibility-based filtering: As explained in R1, DreamGym uses a top-down search over agent-generated trajectories and keeps only those intermediate states that have led to at least one successful terminal outcome, while discarding states whose subsequent rollouts consistently fail. This ensures that only grounded and non-hallucinatory transitions are retained; Meanwhile, this least-feasibility criterion naturally identifies potentially challenging states, i.e., states that are feasible yet difficult, ensuring that the retained transitions remain informative for policy improvement rather than reinforcing erroneous rollouts; (2) Controlled synthetic proportion via the ratio parameter $\beta$: as detailed in R1, we cap the proportion of synthetic transitions stored in the replay buffer. After each epoch, old synthetic entries are discarded and replaced only with new, feasibility-filtered transitions, preventing synthetic data from dominating or drifting the distribution; (3) Reweighted sampling during exploitation: When retrieving examples for next-state prediction, we downweight synthetic transitions once sampled, ensuring a balanced mixture of real and synthetic demonstrations and mitigating the risk of over-reliance on model-generated data.
> >
> > As also shown in the replay-buffer ablation (Table-r. 1 in R1), incorporating synthetic transitions up to 30% does not degrade the quality of the predicted states, where the scores remain identical to the purely real-data setting. These results further validate that our replay buffer update mechanisms effectively prevent low-quality synthetic data from accumulating and successfully prioritize transitions that contribute to meaningful agent policy improvement.
> >
> >
> > Once again, we sincerely thank the reviewer for their thoughtful feedback and appreciation of our work. We hope that our responses have fully addressed your concerns, and we would be grateful if you would consider raising your score if you feel the issues have been satisfactorily resolved. We would also be more than happy to further clarify any remaining questions or concerns. Thank you again for your time and consideration.
> >
> > Best regards,
> >
> > Submission #19123 Authors
> >
> >
> > [1] Li, Y., Inan, H. A., Yue, X., Chen, W. N., Wutschitz, L., Kulkarni, J., ... & Rajmohan, S. (2025). Simulating Environments with Reasoning Models for Agent Training. arXiv preprint arXiv:2511.01824.
> >
> > [2] Wang, Y., Yin, D., Cui, Y., Zheng, R., Li, Z., Lin, Z., ... & Chang, K. W. (2025). LLMs as Scalable, General-Purpose Simulators For Evolving Digital Agent Training. arXiv preprint arXiv:2510.14969.

---

### Official Review · Reviewer_JoEU · 2025-10-31

**Soundness:** 2
**Presentation:** 3
**Contribution:** 1
**Rating:** 2
**Confidence:** 3

**Summary:**

This paper proposes DREAMGYM, a unified and scalable RL training framework for LLM agents. The core idea is to compress environment dynamics into a reasoning-based experience model that operates in an abstract textual state space, producing consistent state transitions and feedback. Coupled with an experience replay buffer (seeded with offline data and continually enriched with online trajectories) and reward-entropy–based curriculum task generation, DREAMGYM enables low-cost, efficient RL training entirely in a “synthetic environment,” while supporting sim-to-real transfer. Experiments show that on the non-RL-ready WebArena, DREAMGYM surpasses baselines by >150%; in RL-ready settings (WebShop/ALFWorld), purely synthetic interactions can match GRPO/PPO; and with sim-to-real, using ≤10% real interactions yields a further 64.5% gain. The paper attributes practical difficulties to costly real-world interactions, insufficient task diversity, unstable reward signals, and heavy infrastructure burdens; DREAMGYM addresses these via a unified abstract state space, vectorized/unified rewards, and an expandable task set.

**Strengths:**

1. By training in synthetic environments and combining a model capable of distilling experience with experience replay and an automatic problem-generation (curriculum) mechanism, the authors reduce dependence on the real system and improve training stability and efficiency. On benchmarks such as WebArena, WebShop, and ALFWorld, the method achieves strong results, which further improve after incorporating a small amount of real data.

2. The exposition is clear: the paper’s narrative is well structured and, overall, easy to read.

**Weaknesses:**

1. The work lacks experimental comparisons with representative methods in the “LLM Agents Reinforcement Learning” line of research; existing comparisons focus mainly on SFT/DPO and PPO/GRPO, making it difficult to accurately define this method’s relative advantages and applicability boundaries among peer approaches. Meanwhile, the related-work survey is neither systematic nor sufficiently comprehensive, with inadequate coverage of recent developments.

2. Although the paper claims reduced real-world interaction and engineering cost, it does not report verifiable metrics (e.g., wall-clock time, GPU hours, throughput/token cost). This prevents a reliable assessment of the return on investment across different compute budgets and scales.

3. Missing pseudocode; key details—such as data filtering/cleaning procedures, prompt templates, random seeds, critical hyperparameter tables and stopping criteria, and the order/frequency of alternating updates between the experience model and the policy—are insufficiently specified, and no accompanying scripts are provided, resulting in a high overall barrier to reproducibility.

**Questions:**

Please refer to the “Weakness” section for related questions.

---

> ### Author Response · Authors · 2025-11-23
>
> Dear Reviewer JoEU,
>
>
> Thank you for acknowledging the novelty of our work and for providing valuable feedback. We have carefully followed your suggestions and made several improvements to the paper, including strengthening the related works section, adding details to enhance reproducibility, providing further clarification on interaction and training costs, and conducting additional experiments against a broader set of baselines and peer methods. These revisions have been incorporated into the paper and are summarized in the responses below. We hope they adequately address your concerns.
>
>
> > W1: The work lacks experimental comparisons with representative methods in the “LLM Agents Reinforcement Learning” line of research; existing comparisons focus mainly on SFT/DPO and PPO/GRPO, making it difficult to accurately define this method’s relative advantages and applicability boundaries among peer approaches. Meanwhile, the related-work survey is neither systematic nor sufficiently comprehensive, with inadequate coverage of recent developments.
>
>
> **R1:** We sincerely appreciate the reviewer for raising this important point, and we fully agree that additional comparisons with representative RL baselines and peer approaches would better contextualize the advantages and unique contribution of our method. Following the reviewer’s suggestion, we have made three major updates:
> (1) we added a comprehensive experimental comparison against representative RL algorithms beyond DPO, PPO, and GRPO;  (2) we added an additional experiment comparing DreamGym with concurrent synthetic-training approaches; and  (3) we substantially revised the related-works section to provide a more systematic discussion of recent developments and to more clearly position our contribution. Below we elaborate on each update.
>
>
>
> ### **Additional comparison with representative RL algorithms**
>
> We compare DreamGym against (1) generic RL baselines including DAPO [1] and Dr. GRPO [2], which are more recent and stable variants of GRPO, and (2) domain-specific RL frameworks including WebAgent-R1 [3], WebRL [4], and DigiRL [5], which require extensive engineering tailored specifically for web environments (and thus cannot transfer to AlfWorld). To provide a fair comparison across diverse scenarios, we evaluate all applicable methods on both WebArena and AlfWorld, reporting average success rate, training time, and number of real-environment trajectories. As shown in Table-r.1 below:
>
> - Consistent to the main results in Table 1 from the paper, DreamGym achieves much higher success rates than generic RL baselines such as DAPO and Dr. GRPO, where they still suffer from the inherent issues of environment infrastructure challenges and low sample efficiency.
> - DreamGym performs comparably to domain-engineered RL frameworks like WebAgent-R1 and WebRL, which rely heavily on real online interactions, despite that DreamGym use no real online trajectories.
> - DreamGym-S2R, after a small amount of continued training in real environments, outperforms WebAgent-R1 in averaged task success rate, which uses substantially more online data and incur significantly higher training overhead.
> - On AlfWorld, DreamGym maintains strong performance, while domain-specific systems cannot transfer, further demonstrating DreamGym’s adaptability to synthesize diverse experience data for arbitrary real-world environments and provide scaleble RL support.

---

> > ### Author Response · Authors · 2025-11-23
> >
> > **Table-r. 1** Comparison of DreamGym against RL algorithms including (1) generic RL baselines (e.g. DAPO and Dr. GRPO) and (2) domain-engineered RL frameworks (e.g. WebAgent-R1, WebRL, DigiRL) on WebArena and AlfWorld. Metrics include averaged success rate (%), estimated time cost (hours), and the total number of real trajectories for training.
> > | Method       | WebArena     |          |                      | AlfWorld     |          |                      |
> > | ---------------- | ---------------- | -------- | -------------------- | ---------------- | -------- | -------------------- |
> > |                  | Success Rate | Time | #Real Traj | Success Rate | Time | #Real Traj |
> > | DAPO         |      10.0            |  120        |      1200 online                |           66.1       |    50      |          4800 online            |
> > | Dr. GRPO     |      12.7            |   120       |     1200 online                 |          70.8        |    50       |     4800 online                  |
> > | WebAgent-R1  |      44.8            |  70        |    2400 online + 9460 offline                  | NA           | NA   | NA               |
> > | WebRL        |     42.4             |   80       |  4000 online + 12200 offline                    | NA           | NA   | NA               |
> > | DigiRL        |     30.3             |  80        |  4000 online + 12200 offline                     | NA           | NA   | NA               |
> > | DreamGym     |      45.0            |   15       |    4800 offline                  |        70.8          |   30        |    5200 offline                    |
> > | DreamGym-S2R |      48.3            |   55       |     800 online + 4800 offline                 |    73.3               |  42       |  1600 online +  5200 offline                  |
> >
> >
> > ### **Additional comparison with peer approaches on synthetic agent training**
> >
> > To further highlight DreamGym’s advantages in model-based RL training for LLM agents, we compare it with two concurrent synthetic-training frameworks: Simia-RL [6] and UI-Simulator [7] (both released later than our first submission). Results are shown in Table-r.2 below and in Appendix C.2 (Table 6) of the revised paper. From the results, we can draw the following conclusion:
> >
> > - Compared to UI-Simulator which synthesizes the complete raw DOM trees for full-fidelity simulation, DreamGym achieves substantially higher success rates and significantly lower time costs. This advantage stems from DreamGym simulating transitions in a concise, explicit, and token-efficient abstract state space, which greatly reduces environment inference overhead and yields more coherent and policy-relevant transitions.
> >
> > - Compared to Simia-RL which relies on a closed-source SOTA model (e.g. OpenAI o4-mini) , DreamGym reduces monetary cost by over 30× by using a fine-tuned open-source simulator (see the detailed annotation cost analysis in our response to W2). DreamGym also achieves higher success rates, as fine-tuning on domain-specific offline trajectories produces more consistent, in-distribution, and reliable transitions that directly support policy improvement.
> >
> > ---
> >
> > **Table-r. 2** Comparison of DreamGym against concurrent synthetic training frameworks on the WebArena environment. Metrics include averaged success rate (%), estimated time cost (hours), number of real-world trajectories used for training, and the monetary expense of environment service (USD) such as AWS server hosts and API key costs.
> > | Method | Success Rate |Time | #Real Traj| Expense|
> > |--------|---------|----------|-----------|-----------|
> > | Traditional       |15.7 |120 | 1200 online | 272 |
> > | Simia-RL    | 37.0 | 17 |4800 offline  |460.8 |
> > | UI-Simulator |19.7 | 22 | 4800 offline | 72 |
> > | DreamGym    |45.0 | 15 | 4800 offline | 14.4 |
> >
> >
> > Therefore, by modeling the environment in a learning-oriented, token-efficient meta-representation space, DreamGym is significantly more cost-effective and produces experience data that directly supports policy improvement without incurring unnecessary fidelity-related overhead.

---

> > > ### Author Response · Authors · 2025-11-23
> > >
> > > ### **Revisions to the related-works survey**
> > >
> > > We have thoroughly revised the related-works section to address the reviewer’s concern such that it is sufficiently systematic and has comprehensive coverage of recent developments.
> > >
> > > In Section 2.1 (_LLM Agents Reinforcement Learning_), we now:
> > > - systematically discuss recent RL paradigms for LLM training, covering both preference-based alignment and verifiable RL for improving mathematical reasoning and coding capabilities, and we detail specific algorithms including DPO, PPO, GRPO, DAPO, and Dr. GRPO;
> > > - identify key challenges of applying these algorithms to long-horizon agent tasks such as limited task sets, sparse rewards, and infrastructure instability;
> > > - furthermore, we introduce more recent domain-specific RL frameworks such as WebAgent-R1 and WebRL that devote extensive engineering efforts to bridge the infrastructure gap for training RL agents in WebArena (e.g., reset reliability, data-collection scalability);
> > > - however, these approaches remain fundamentally constrained by real-world environment dynamics, motivating DreamGym as a **scalable, unified RL framework** capable of transferring across environments without overwhelming engineering effort.
> > >
> > > In Section 2.2 (_Training Agents with Synthetic Data_), we have also expanded our coverage to compare with concurrent synthetic data approaches for training agents:
> > > - discussing early works that rely on direct scripting or mined tutorials to synthesize expert trajectories (primarily for SFT) for policy distillation,
> > > - outlining methods based on synthetic environments for model-based RL training such as Dreamer, and
> > > - conducting an in-depth comparison with two concurrent synthetic-training works, i.e., Simia-RL [6] and UI-Simulator [7] (both released after our initial submission; to the best of our knowledge, DreamGym is the earliest work to propose a complete, scalable framework for training LLM agents in synthetic environments to bridge the RL gap for arbitrary real-world environments).
> > >
> > >
> > >
> > > We highlight that UI-Simulator directly synthesizes raw DOM observations, which is expensive and challenging, while Simia-RL relies on closed-source models and produces costly, potentially out-of-domain data that can only be used for the current online training stage that will be discarded afterward. In contrast, DreamGym uses a low-cost open-source model for annotation and trains a reusable experience model that synthesizes transitions in an abstract meta-representation space that is both learning-oriented for policy improvement and token-efficient for stable simulation, enabling the generation of unlimited, reliable online rollouts for training any agent with RL.
> > >
> > >
> > >
> > > We sincerely hope that the expanded experiments and the substantially improved literature survey address the reviewer’s concerns and help clarify the novelty, generality, and unique contribution of our work.

---

> > > > ### Author Response · Authors · 2025-11-23
> > > >
> > > > > W2: Although the paper claims reduced real-world interaction and engineering cost, it does not report verifiable metrics (e.g., wall-clock time, GPU hours, throughput/token cost). This prevents a reliable assessment of the return on investment across different compute budgets and scales.
> > > >
> > > > R2: We sincerely appreciate the reviewer’s suggestion and have therefore added additional statistics and a more comprehensive discussion of both engineering and monetary costs to clarify the efficiency gains of our work. Specifically, we provide a detailed breakdown of DreamGym’s costs across two main stages, i.e., training data annotation for experience models and agent RL data collection.
> > > >
> > > > ### **Detailed statistics of costs during the data annotation stage for training experience models**
> > > >
> > > > During training, the primary additional cost beyond standard RL pipelines comes from annotating the reasoning traces and task-variation sets using a teacher model. Table-r. 3 below summarizes the annotation costs for WebShop, ALFWorld, and WebArena, including both reasoning-trace and task-variation annotations. We first clarify that all reasoning-trace annotations in DreamGym are generated using Llama-3.3-70B-Instruct, an open-source model that can be served on a single A100 GPU; for a straightforward quantitative estimate, we convert its usage into an equivalent market price of 1 USD per 1M tokens. In particular, we do not rely on expensive closed-source LLMs such as OpenAI o4-mini, which concurrent approaches like Simia-RL [6] heavily depend on for generating online interactions.
> > > >
> > > >
> > > > To annotate the reasoning trace for each trajectory, we provide the LLM with the full trajectory and prompt it to generate a reasoning explanation for each transition step (averaging ~100 tokens per step; see Appendix E for the prompt template). Consequently, the annotation cost scales with the trajectory length (e.g., ALFWorld incurs slightly higher cost due to its 10–20 step average sequence length). Regardless, as shown in Table-r. 3, the overall annotation cost for DreamGym remains low across all environments, requiring less than $10 in total to annotate all data needed to train the experience models.
> > > >
> > > >
> > > > In contrast, concurrent approaches such as Simia-RL incur substantial ongoing costs, since they rely on SOTA models like OpenAI o4-mini to interact with the agent online throughout training, where this process must be repeated every time a new agent is trained. As shown in Table-r. 2 above, even a modest training configuration (e.g., 150 epochs with batch size 16) results in significant inference expenses, amounting to over 30× the cost of DreamGym. On the contrary, DreamGym avoids this repeated cost by design: annotation is performed _once-and-for-all_ in an offline setting, and the resulting fine-tuned experience model can then be reused indefinitely to generate online interactions for any agent without incurring additional inference expenses.
> > > >
> > > >
> > > > To further address the reviewer’s concern, we also conduct an ablation comparing Llama-3.3-70B with GPT-4.1 as the annotating model, as shown in Table-r. 4 below. GPT-4.1 provides only a small performance improvement over Llama-3.3-70B while costing 5× more, indicating a relatively low marginal gain. This suggests that the reasoning-trace annotation stage does not require LLMs with extremely strong reasoning capability and a powerful open-source model like Llama-3.3-70B is sufficient for the task. Accordingly, we use Llama-3.3-70B to annotate all datasets in our experiments.
> > > >
> > > >
> > > > ---
> > > >
> > > > **Table-r. 3** Detailed annotation statistics. We report the number of trajectories, reasoning-trace annotation cost (USD), and task-variation annotation cost (USD) for WebShop, ALFWorld, and WebArena. Although Llama-3.3-70B-Instruct was served locally, we estimate cost using a market rate of \$1 per 1M tokens for comparability.
> > > >
> > > > | Method                         | WebShop | AlfWorld | WebArena |
> > > > |-------------------------------|---------|----------|----------|
> > > > | #Trajectory                   | 3600    | 5200     | 4800     |
> > > > | Reasoning Annotation Costs    | 3.6     | 7.8      | 7.2      |
> > > > | Task Variation Annotation Costs | 0.5   | 0.5      | 0.5      |
> > > >
> > > > ---
> > > >
> > > > **Table-r. 4** Comparison of annotation costs (USD) and averaged success rate when using Llama-3.3-70B vs. GPT-4.1 as the reasoning-trace annotator.
> > > >
> > > > | Method             | Llama-3.3-70B | GPT-4.1 |
> > > > |--------------------|---------------|---------|
> > > > | Annotation Costs   | 7.2           | 36      |
> > > > | Success Rate       | 45.0          | 45.6    |

---

> > > > > ### Author Response · Authors · 2025-11-23
> > > > >
> > > > > ### **Detailed statistics of costs during the agent RL data collection stage**
> > > > >
> > > > > Following the reviewer’s suggestion, we also provide detailed statistics of the costs incurred during the agent inference stage (i.e., RL data collection). Table-r.5 reports the average wall-clock time during the experience model generation per trajectory, the average number of tokens produced by experience model per trajectory, and the total wall-clock training time across WebShop, ALFWorld, and WebArena. In addition, we include a head-to-head comparison between training agents in DreamGym and training in the real-world environment on WebArena, covering several verifiable metrics, including the average time during experience model generation per step, average time during experience model generation per trajectory, environment reset cost, and total training time, as summarized in Table-r.6 below.
> > > > >
> > > > > **Per-step inference:** To accommodate fast environment inference, we serve the 8B experience model using an efficient vLLM inference stack on 2×A100 GPUs. As shown in both Table-r.5 and Table-r.6, this results in very low inference latency in practice: DreamGym requires only 0.08 seconds on average to generate the reasoning trace and next-state transition for a single step. Importantly, DreamGym consumes **less than one second in total** to generate all environment states for an entire trajectory, which is negligible compared to overall GPU training time. It also consumes **fewer than 10K tokens per trajectory** (approximately 300–600 tokens per step), since the environment is synthesized in a token-efficient and sufficiently expressive abstract state space. In contrast, the real WebArena environment, even when following the official guidelines and hosting 8 concurrent instances on AWS to minimize the engineering gap still takes **3.4 seconds per step** due to substantial overhead such as HTTP requests, page rendering, and resource fetching. This does not include the additional human intervention often required to resolve issues such as pages getting stuck on login screens or unstable server responses, all of which further degrade the speed and reliability of real-environment execution.
> > > > >
> > > > >
> > > > > **Environment reset**: Moreover, as recommended by WebArena’s official documentation, we manually reset all environment Dockers after every 50 tasks to avoid cross-task interference, where each reset takes 300 seconds on average. Although automated scripts are possible, we found them unstable and less operationable in practice. On the contrary, DreamGym does not incur any comparable overhead and resetting the environment simply involves resetting the prompt, which is instantaneous.
> > > > >
> > > > > **Sampling failures**: In addition, we also report the failure rate during trajectory sampling, which is the proportion of trajectories where at least one state transition fails. Such failures frequently occur in WebArena due to infrastructure instability, e.g., network timeouts, page breakdowns, and cross-task interference, an open issue also explicitly acknowledged in their repo. In contrast, while DreamGym also incur certain failures (e.g., occasional repetition patterns in model outputs), they occur far less frequently as model-based transitions are more reliable and controllable. And importantly, unlike the real environment where a single failed transition invalidates the entire trajectory (requiring resampling from the beginning), DreamGym can simply resample from the failed state, significantly improving efficiency and fault tolerance.
> > > > >
> > > > > Eventually, these factors accumulate into large differences in overall training time. As shown in Table-r.4, DreamGym completes a full training run in 15 hours, with the majority of time spent on GPU training rather than environment inference. In contrast, training with the real WebArena environment requires over 120 hours, where environment interaction dominates the total cost. Moreover, simply hosting the WebArena instances on AWS incurs a non-negligible monetary expense (approximately \$272 for 120 hours), as reported in Table-r.1 above.
> > > > >
> > > > >
> > > > > **Table-r. 5** Detailed statistics of costs during agent RL data collection across WebShop, AlfWorld, and WebArena environments, including wall-clock training time (hours), average tokens generated by the experience model per trajectory, and wall-clock time during experience model generation per trajectory (seconds).
> > > > >
> > > > > | Metric                         | WebShop | AlfWorld | WebArena |
> > > > > |--------------------------------|---------|----------|----------|
> > > > > | Wall-clock training time (h)   | 12      | 19       | 15       |
> > > > > | Avg. tokens per trajectory     | 2k      | 5k       | 8k       |
> > > > > | Wall-clock time per trajectory (s) | 0.3  | 0.5      | 0.9      |

---

> ### Author Response · Authors · 2025-11-23
>
> **Table-r. 6** Detailed breakdown of time costs on WebArena, including average time per step (seconds), average time per trajectory (seconds), environment reset time (seconds), total training time (hours), and trajectory sampling failure rate.
>
> | Method           | Avg Time per Step (s) | Avg Time per Traj (s) | Reset Time (s) | Total Training Time (h) | Sampling Failure Rate (%) |
> |------------------|------------------------|-------------------------|----------------|---------------------------|----------------------------|
> | DreamGym         | 0.08                   | 0.9                     | 0              | 15                        | 1.7                        |
> | Real Environment | 3.4                    | 54.4                    | 300            | 120                       | 19.0                       |
>
>
>
> We hope these clarifications address the reviewer’s suggestion for a more detailed discussion of the verifiable cost metrics across the different stages of DreamGym and provide a clearer, more reliable assessment of the return on investment under varying compute budgets and training scales.
>
>
> > W3: Missing pseudocode; key details—such as data filtering/cleaning procedures, prompt templates, random seeds, critical hyperparameter tables and stopping criteria, and the order/frequency of alternating updates between the experience model and the policy—are insufficiently specified, and no accompanying scripts are provided, resulting in a high overall barrier to reproducibility.
>
>
> R3: We deeply appreciate the reviewer's suggestion and fully agree that providing additional methodological and experimental details strengthens the presentation of our work, clarifies the unique contributions of our approach, and greatly improves reproducibility. In the revision, we have added the following key components: (1) detailed pseudocode formalizing the full experience-synthesis and RL training pipeline of DreamGym; (2) an expanded explanation of the data filtering and cleaning procedures for each environment; (3) a complete list of all prompt templates used in our framework; and (4) a comprehensive hyperparameter table including training configurations and random seeds.
>
> **Detailed Pseudocode for Experience Synthesis and Training Pipeline**:
>
> Concretely, we provide detailed pseudocode in Appendix C.1 to formalize the data synthesis procedure and RL training pipeline in DreamGym, covering all critical steps, including:
> (i) initialization of all components, including the agent policy (initialized from a base model), the replay buffer (seeded with offline trajectories), and the task pool (initialized with a set of seed tasks);
> (ii) synthesizing multi-turn rollouts by iteratively querying the reasoning-based experience model using the current state–action pair, task instruction, interaction history, and top-$k$ retrieved experiences;
> (iii) updating the agent policy with an RL optimizer using the collected synthetic experience data; and
> (iv) computing group-based reward entropy to drive curriculum task generation and updating both the task pool and replay buffer throughout training.

---

> > ### Author Response · Authors · 2025-11-23
> >
> > **Expanded explanation of the data filtering and cleaning procedures:**
> > DreamGym involves two major stages for training RL agents: (1) training a reasoning-based experience model in the target environment to synthesize next states and rewards via explicit reasoning, and (2) interacting with this experience model to collect abundant, diverse online experience for agent RL training. Since stage (2) follows standard procedures shared by most RL frameworks and is agnostic to the downstream RL algorithm, we focus here to elaborate on the data filtering and quality-control procedures used in stage (1).
> >
> > For each environment, our data filtering pipeline consists of the following three steps:
> >
> > (i) Collecting offline trajectories from multiple sources.
> > For example, in WebArena, we gather trajectories from various publicly available sources, such as the top-10 agent trace submissions on the official WebArena leaderboard, as well as the oracle trajectories in the WebArena-Lite repository, which yield more than 10K raw trajectories. Notably, we retain both successful and failed trajectories, as both encode the environment dynamics and are valuable for distilling the transition structure into the experience model. In this step, we also carefully remove all tasks and trajectories corresponding to the test set to avoid any form of test contamination.
> >
> > (ii) Balancing trajectories across available training tasks as well as success and failure modes.
> > To reduce bias during experience-model training, we first balance the trajectories collected in step (i) to ensure relatively even coverage across all training tasks. We then further filter and rebalance the remaining trajectories so that each task contains a more uniform mix of successful and failed trajectories. This mitigates overfitting to dominant failure patterns (e.g., predicting success even when actions previously failed) and enables the experience model to learn more reliable and robust environment transitions.
> >
> >
> > (iii) Ensuring diverse coverage of the action space.
> > Based on the trajectories from the previous step, we then apply a clustering-based selection procedure to maintain coverage over diverse action sequences. This avoids overfitting the experience model to a narrow subset of actions and encourages it to learn transitions for a broad range of behaviors and accurately predict their consequences.
> >
> > After applying these three stages, we obtain 3600, 5200, and 2780 high-quality trajectories for WebShop, ALFWorld, and WebArena, respectively. We then use Llama-3.3-70B as the annotator to generate a reasoning trace for each state transition in the filtered dataset, which serves as the training corpus for fine-tuning the experience model.
> >
> > Overall, this simple yet effective data filtering pipeline allows us to efficiently collect a large-scale offline trajectories from various sources and transform them into a diverse, balanced, and clean dataset well-suited for training a stable and reliable experience model, which can subsequently generate high-quality interaction data for RL agent training.
> >
> >
> >
> > **Complete list of all prompt templates:**
> > In our first submission, we already included a complete list of all prompt templates in Appendix F for every environment used in our experiments. These templates cover (i) prompts used during the data-annotation stage, including initial data filtering, reasoning-trace annotation, and task-variation generation; (ii) prompts used during the experience-model inference stage, such as next-state prediction and new-task generation; and (iii) prompts used for agent policy inference in the corresponding environments. Appendix F presents every prompt exactly as used in our implementation, along with environment-specific variations where applicable, to ensure full transparency and reproducibility. We hope these additions clarify the methodological details and enable the reviewer and future researchers to fully validate and build upon our framework.

---

> > > ### Author Response · Authors · 2025-11-23
> > >
> > > **Comprehensive hyperparameter and configurations:**
> > >
> > > While we have already provided all the necessary hyperparameters and configurations for our experiments in Section 5 and Appendix A, we fully agree with the reviewer that a more detailed summary table would further improve reproducibility. In the revised version of the paper, we now include two tables: Table 3 in Appendix B.1 summarizes the hyperparameters for the experience model training and inference, including the annotation model, random seed, top-$k$ examples, synthetic task ratio, temperature, and whether outcome-based supervision is enabled. And similarly, Table 4 in Appendix B.1 summarizes all hyperparameters for agent RL training, including random seed, batch size, mini-batch size, micro-batch size, KL-loss coefficient, and learning rate. Specifically, all SFT hyperparameters follow the best practices recommended by the LLaMA-Factory repository, and all RL hyperparameters follow the default settings from Verl-Agent; we do not introduce additional tricks or perform deliberate hyperparameter tuning.
> > >
> > > We believe these expanded details on hyperparameter configurations will make our experiments easier to reproduce and provide a clearer foundation for future work to extend or build upon our framework.
> > >
> > > We truly thank the reviewer for the valuable feedback and suggestions that helped improve our work. We hope that our responses have fully addressed your concerns, and we would be grateful if you would consider reevaluating your score so that we can share this work with the broader community. Thank you very much for your time and effort!
> > >
> > > Best regards,
> > >
> > > Submission #19123 Authors
> > >
> > >
> > > [1] Yu, Q., Zhang, Z., Zhu, R., Yuan, Y., Zuo, X., Yue, Y., ... & Wang, M. (2025). Dapo: An open-source llm reinforcement learning system at scale. arXiv preprint arXiv:2503.14476.
> > >
> > > [2] Liu, Z., Chen, C., Li, W., Qi, P., Pang, T., Du, C., ... & Lin, M. (2025). Understanding r1-zero-like training: A critical perspective. arXiv preprint arXiv:2503.20783.
> > >
> > > [3] Wei, Z., Yao, W., Liu, Y., Zhang, W., Lu, Q., Qiu, L., ... & Li, L. (2025). Webagent-r1: Training web agents via end-to-end multi-turn reinforcement learning. arXiv preprint arXiv:2505.16421.
> > >
> > > [4] Qi, Z., Liu, X., Iong, I. L., Lai, H., Sun, X., Zhao, W., ... & Dong, Y. (2024). Webrl: Training llm web agents via self-evolving online curriculum reinforcement learning. arXiv preprint arXiv:2411.02337.
> > >
> > > [5] Bai, H., Zhou, Y., Pan, J., Cemri, M., Suhr, A., Levine, S., & Kumar, A. (2024). Digirl: Training in-the-wild device-control agents with autonomous reinforcement learning. Advances in Neural Information Processing Systems, 37, 12461-12495.
> > >
> > > [6] Li, Y., Inan, H. A., Yue, X., Chen, W. N., Wutschitz, L., Kulkarni, J., ... & Rajmohan, S. (2025). Simulating Environments with Reasoning Models for Agent Training. arXiv preprint arXiv:2511.01824.
> > >
> > > [7] Wang, Y., Yin, D., Cui, Y., Zheng, R., Li, Z., Lin, Z., ... & Chang, K. W. (2025). LLMs as Scalable, General-Purpose Simulators For Evolving Digital Agent Training. arXiv preprint arXiv:2510.14969.

---

### Official Review · Reviewer_D1t6 · 2025-11-09

**Soundness:** 4
**Presentation:** 3
**Contribution:** 3
**Rating:** 6
**Confidence:** 3

**Summary:**

This paper tackles the significant challenges of applying reinforcement learning RL to LLM agents, namely the high cost of environment interaction, unreliable reward signals, and the difficulty of scaling task diversity . The authors propose DreamGYM, a framework that synthesizes agent experiences to enable effective and scalable online RL. Instead of interacting with a costly real environment, the agent interacts with a "reasoning-based experience model". This model, an LLM trained on offline trajectories annotated with CoT reasoning , learns to predict state transitions and feedback signals in an abstract textual space. The framework has two other key components: an experience replay buffer (seeded with offline data and enriched with new interactions) to stabilize training, and an adaptive curriculum task generator that uses a "reward entropy" heuristic to create progressively more challenging tasks for the agent. The experiment show that the purely synthetic approach is competitive with baselines trained on 80K real interactions. On the "non-RL-ready" WebArena benchmark, DreamGYM enables RL training and achieves a relative improvement of over 150% compared to baselines. The sim-to-real approach demonstrates significant performance gains and sample efficiency.

**Strengths:**

The paper targets an important bottleneck in the field of autonomous agents. The prohibitive cost and low sample efficiency of online RL in complex, real-world environments (like web browsers) is a major barrier to progress. The paper's goal of creating a scalable, synthetic training environment is highly relevant and impactful.

The framework's ability to enable effective RL on WebArena, an environment considered "not RL-ready", is the most compelling result. Achieving success rates over 30-45% where previous methods (including traditional RL) struggle to get past 10-19%  is a significant practical achievement.

**Weaknesses:**

The central idea is to use an LLM as a learned world model for model-based RL. The paper cites related work like Dreamer and other LLM-based environment models but claims to be different by focusing on "policy improvement" rather than "fidelity-first". This is a weak distinction, as policy improvement is the goal of all MBRL systems, including Dreamer. The main novelties are the (effective) use of CoT reasoning in the model's SFT training and the entropy-based curriculum generator. These are good contributions, but the overall framework is a (well-executed) application of established MBRL concepts to the LLM agent domain, not a fundamental new paradigm.

Besides, the paper emphasizes that it avoids costly real-world rollouts. However, it includes two important, undiscussed costs:
- Annotation: The experience model is trained on thousands of trajectories (e.g., 3.6K for WebShop, 5.2K for ALFWorld ), each annotated with reasoning traces by a "powerful LLM". This is a substantial one-time computational cost, relying on a (presumably) proprietary, high-capacity model (like GPT-4) that is not part of the open-source setup.
- Inference: The "fast" synthetic environment is itself a Llama-3.1-8B model. Generating each step of a trajectory requires a full inference pass from this 8B model . While this avoids infrastructure issues like Docker, it is still computationally intensive. Figure 3 (Left)  shows a reduction in total training time, but a clearer analysis of the wall-clock time per step (Real Env vs. DreamGYM) is needed to fully assess the *efficiency* claim.

**Questions:**

Please refer to the weakness part above.

---

> ### Author Response · Authors · 2025-11-23
>
> Dear Reviewer D1t6,
>
> Thank you for your appreciation of the value and novelty of our proposed method! We have carefully followed your suggestions, added further clarification on the costs during the annotation and inference stage, and conducted additional experiments against concurrent MBRL-based baselines. These revisions have been incorporated into the paper and are summarized in the responses below. We hope they adequately address your concerns.
>
>
> > W1: The central idea is to use an LLM as a learned world model for model-based RL. The paper cites related work like Dreamer and other LLM-based environment models but claims to be different by focusing on "policy improvement" rather than "fidelity-first". This is a weak distinction, as policy improvement is the goal of all MBRL systems, including Dreamer. The main novelties are the (effective) use of CoT reasoning in the model's SFT training and the entropy-based curriculum generator. These are good contributions, but the overall framework is a (well-executed) application of established MBRL concepts to the LLM agent domain, not a fundamental new paradigm.
>
> **R1:** We thank the reviewer for recognizing the effectiveness and novelty of our framework. While we completely agree with the reviewer that policy improvement is the core objective of all MBRL systems, our key contribution lies in showing that **training within an abstract, meta-representation space that prioritizes high-entropy and diverse experiences, is substantially more effective and more scalable for policy improvement** than approaches that primarily target environment fidelity, which are both costly and difficult to scale.
>
> We also agree that our approach is grounded in established MBRL principles. Our goal is not to redefine MBRL, but rather to address several fundamental challenges in transferring the MBRL paradigm to the LLM-agent setting, including (i) highly complex dynamics of diverse real-world environments, (ii) limited diversity in model-based state prediction, and (iii) prohibitive inference cost of fidelity-oriented simulation. To tackle these issues, our framework introduces a reasoning-based experience model that synthesizes transitions in a concise and token-efficient representation space, paired with a grounded replay buffer to control hallucination and maintain distributional diversity. In addition, DreamGym employs an entropy-based curriculum task generator to provide high-entropy, informative experiences that further enhance policy improvement.
>
> To further demonstrate the benefit of our design for policy improvement, we compare DreamGym with concurrent MBRL-based agent training frameworks such as Simia-RL [1], which uses OpenAI o4-mini to replicate full environment feedback, and UI-Simulator [2], which directly synthesizes raw DOM observations, both of which follow a “fidelity-first’’ simulation strategy. Results are presented in Table-r. 1 below and also in Table 6 of Appendix C.2 in the revised paper. Specifically:
>
>
> - Compared to UI-Simulator, which generates complete, raw DOM trees, DreamGym achieves substantially higher success rates and lower time costs. This stems from simulating in a concise and well-structured abstract state space, significantly reducing environment inference overhead and yielding more coherent and informative transitions for policy improvement.
>
> - Compared to Simia-RL, which relies on a closed-source SOTA model for full-fidelity simulation, DreamGym reduces monetary cost by more than 30× by instead using a simple open-source simulator (see our detailed annotation-cost comparison in the response to W2). In addition, DreamGym achieves higher success rates as we fine-tune the simulator on domain-specific offline data, yielding more coherent, in-distribution state transitions that are directly conducive to policy improvement.
>
> ---
>
> **Table-r. 1** Comparison of DreamGym against concurrent synthetic training frameworks on the WebArena environment. Metrics include averaged success rate (%), estimated time cost (hours), number of real-world trajectories used for training, and the monetary expense of environment service (USD) such as AWS server hosts and API key costs.
> | Method | Success Rate |Time | #Real Traj| Expense|
> |--------|---------|----------|-----------|-----------|
> | Traditional       |15.7 |120 | 1200 online | 272 |
> | Simia-RL    | 37.0 | 17 |4800 offline  |460.8 |
> | UI-Simulator |19.7 | 22 | 4800 offline | 72 |
> | DreamGym    |45.0 | 15 | 4800 offline | 14.4 |
>
>
> ---
>
> In summary, by modeling the environment in a more informative, token-efficient abstract space that is learning-oriented, DreamGym is significantly more cost-effective and produces experience data that directly supports policy improvement without incurring unnecessary fidelity-related overhead. We hope this clarifies the novelty and motivation of our approach and addresses your concerns.

---

> > ### Author Response · Authors · 2025-11-23
> >
> > > W2: Annotation Costs: The experience model is trained on thousands of trajectories (e.g., 3.6K for WebShop, 5.2K for ALFWorld ), each annotated with reasoning traces by a "powerful LLM". This is a substantial one-time computational cost, relying on a (presumably) proprietary, high-capacity model (like GPT-4) that is not part of the open-source setup.
> >
> >
> > **R2:** We thank the reviewer for raising this important question. Following the reviewer’s suggestion, we now provide a detailed breakdown of the annotation costs across all stages for all the benchmark environments in Table-r. 2 below. First, we clarify that all reasoning-trace annotations in DreamGym are generated using Llama-3.3-70B-Instruct, an open-source model that can be served on a single A100 GPU, and we do not rely on closed-source LLMs such as OpenAI o4-mini, which Simia-RL [1] significantly employs to generate online interactions.
> >
> > To be specific, Table-r. 2 summarizes the cost of reasoning-trace annotations and task-variation annotations for WebShop, ALFWorld, and WebArena. For each trajectory, the LLM receives the full trajectory and generates a reasoning annotation for each transition step (on average ~100 tokens per step; see Appendix E for the prompt template). Thus, annotation cost scales with the average trajectory length (e.g., ALFWorld has longer sequences of 10–20 steps). As shown, the overall annotation cost is low across all environments.
> >
> > In contrast, concurrent approaches such as Simia-RL incur substantial *ongoing* costs, since they rely on SOTA models like OpenAI o4-mini to interact with the agent online throughout training, where this process must be repeated every time a new agent is trained. As shown in Table-r. 1 above, even a modest training configuration (e.g., 150 epochs with batch size 16) results in significant inference expenses, amounting to over 30× the cost of DreamGym. As the reviewer also mentioned, DreamGym avoids this repeated cost by design: annotation is performed _once-and-for-all_ in an offline setting, and the resulting fine-tuned experience model can then be reused indefinitely to generate online interactions for any agent without incurring additional inference expenses.
> >
> >
> > To further address the reviewer’s concern, we also conduct an ablation comparing Llama-3.3-70B with GPT-4.1 as the annotating model (Table-r. 3). GPT-4.1 provides only a small performance improvement over Llama-3.3-70B while costing 5× more, indicating a relatively low marginal gain. This suggests that the reasoning-trace annotation stage does not require LLMs with extremely strong reasoning capability and a powerful open-source model like Llama-3.3-70B is sufficient for the task. Accordingly, we use Llama-3.3-70B to annotate all datasets in our experiments.
> >
> >
> > ---
> >
> > **Table-r. 2.** Detailed annotation statistics. We report the number of trajectories, reasoning-trace annotation cost (USD), and task-variation annotation cost (USD) for WebShop, ALFWorld, and WebArena. Although Llama-3.3-70B-Instruct was served locally, we estimate cost using a market rate of \$1 per 1M tokens for comparability.
> >
> > | Method                         | WebShop | AlfWorld | WebArena |
> > |-------------------------------|---------|----------|----------|
> > | #Trajectory                   | 3600    | 5200     | 4800     |
> > | Reasoning Annotation Costs    | 3.6     | 7.8      | 7.2      |
> > | Task Variation Annotation Costs | 0.5   | 0.5      | 0.5      |
> >
> >
> > **Table-r. 3.** Comparison of annotation costs (USD) and averaged success rate when using Llama-3.3-70B vs. GPT-4.1 as the reasoning-trace annotator.
> >
> > | Method             | Llama-3.3-70B | GPT-4.1 |
> > |--------------------|---------------|---------|
> > | Annotation Costs   | 7.2           | 36      |
> > | Success Rate       | 45.0          | 45.6    |

---

> > > ### Author Response · Authors · 2025-11-23
> > >
> > > > W3: Inference Costs: The "fast" synthetic environment is itself a Llama-3.1-8B model. Generating each step of a trajectory requires a full inference pass from this 8B model . While this avoids infrastructure issues like Docker, it is still computationally intensive. Figure 3 (Left) shows a reduction in total training time, but a clearer analysis of the wall-clock time per step (Real Env vs. DreamGYM) is needed to fully assess the efficiency claim.
> > >
> > >
> > > **R3:** We completely agree with the reviewer that a more detailed analysis of inference costs would strengthen the efficiency claims of our method. Following the reviewer’s suggestion, we provide a more detailed statistics of the costs incurred during the agent inference stage (i.e., RL data collection). Specifically, Table-r.4 reports the average wall-clock time during the experience model generation per trajectory, the average number of tokens produced by experience model per trajectory, and the total wall-clock training time across WebShop, ALFWorld, and WebArena. In addition, we include a head-to-head comparison between training agents in DreamGym and training in the real-world environment on WebArena, covering several verifiable metrics, including the average time during experience model generation per step, average time during experience model generation per trajectory, environment reset cost, and total training time, as summarized in Table-r.5 below.
> > >
> > > **Per-step inference:** To accommodate fast environment inference, we serve the 8B experience model using an efficient vLLM inference stack on 2×A100 GPUs. As shown in both Table-r.4 and Table-r.5, this results in very low inference latency in practice: DreamGym requires only 0.08 seconds on average to generate the reasoning trace and next-state transition for a single step. Importantly, DreamGym consumes **less than one second in total** to generate all environment states for an entire trajectory, which is negligible compared to overall GPU training time. It also consumes **fewer than 10K tokens per trajectory** (approximately 300–600 tokens per step), since the environment is synthesized in a token-efficient and sufficiently expressive abstract state space. In contrast, the real WebArena environment, even when following the official guidelines and hosting 8 concurrent instances on AWS to minimize the engineering gap still takes **3.4 seconds per step** due to substantial overhead such as HTTP requests, page rendering, and resource fetching. This does not include the additional human intervention often required to resolve issues such as pages getting stuck on login screens or unstable server responses, all of which further degrade the speed and reliability of real-environment execution.
> > >
> > >
> > > **Environment reset**: Moreover, as recommended by WebArena’s official documentation, we manually reset all environment Dockers after every 50 tasks to avoid cross-task interference, where each reset takes 300 seconds on average. Although automated scripts are possible, we found them unstable and less operable in practice. On the contrary, DreamGym does not incur any comparable overhead, and resetting the environment involves resetting the prompt, which is instantaneous.
> > >
> > > **Sampling failures**: Additionally, we report the failure rate during trajectory sampling, which is the proportion of trajectories where at least one state transition fails. Such failures frequently occur in WebArena due to infrastructure instability, e.g., network timeouts, page breakdowns, and cross-task interference, an open issue also explicitly acknowledged in their repo. In contrast, while DreamGym also incurs certain failures (e.g., occasional repetition patterns in model outputs), they occur far less frequently as model-based transitions are more reliable and controllable. And importantly, unlike the real environment, where a single failed transition invalidates the entire trajectory (requiring resampling from the beginning), DreamGym can simply resample from the failed state, significantly improving efficiency and fault tolerance.
> > >
> > > Eventually, these factors accumulate into large differences in overall training time. As shown in Table-r.5, DreamGym completes a full training run in 15 hours, with the majority of time spent on GPU training rather than environment inference. In contrast, training with the real WebArena environment requires over 120 hours, where environment interaction dominates the total cost. Moreover, simply hosting the WebArena instances on AWS incurs a non-negligible monetary expense (approximately \$272 for 120 hours), as reported in Table-r.1 above.

---

> > > > ### Author Response · Authors · 2025-11-23
> > > >
> > > > **Table-r.4.** Detailed statistics of costs during agent RL data collection across WebShop, AlfWorld, and WebArena environments, including wall-clock training time (hours), average tokens generated by the experience model per trajectory, and wall-clock time during experience model generation per trajectory (seconds).
> > > >
> > > > | Metric                         | WebShop | AlfWorld | WebArena |
> > > > |--------------------------------|---------|----------|----------|
> > > > | Wall-clock training time (h)   | 12      | 19       | 15       |
> > > > | Avg. tokens per trajectory     | 2k      | 5k       | 8k       |
> > > > | Wall-clock time per trajectory (s) | 0.3  | 0.5      | 0.9      |
> > > >
> > > >
> > > >
> > > >
> > > > **Table-r. 5.** Detailed breakdown of time costs on WebArena, including average time per step (seconds), average time per trajectory (seconds), environment reset time (seconds), total training time (hours), and trajectory sampling failure rate.
> > > >
> > > > | Method           | Avg Time per Step (s) | Avg Time per Traj (s) | Reset Time (s) | Total Training Time (h) | Sampling Failure Rate (%) |
> > > > |------------------|------------------------|-------------------------|----------------|---------------------------|----------------------------|
> > > > | DreamGym         | 0.08                   | 0.9                     | 0              | 15                        | 1.7                        |
> > > > | Real Environment | 3.4                    | 54.4                    | 300            | 120                       | 19.0                       |
> > > >
> > > >
> > > >
> > > > ---
> > > >
> > > > We hope these clarifications address the reviewer’s request for a more transparent comparison of inference costs between DreamGym and real-environment execution.
> > > >
> > > > Once again, we sincerely thank the reviewer for their appreciation of our work and their constructive feedback. We hope that our responses have fully addressed your concerns, and we would be grateful if you would consider raising your score so we can share this work with the broader community. Thank you for your time and effort!
> > > >
> > > > Best regards,
> > > >
> > > > Submission #19123 Authors
> > > >
> > > >
> > > >
> > > > [1] Li, Y., Inan, H. A., Yue, X., Chen, W. N., Wutschitz, L., Kulkarni, J., ... & Rajmohan, S. (2025). Simulating Environments with Reasoning Models for Agent Training. arXiv preprint arXiv:2511.01824.
> > > >
> > > > [2] Wang, Y., Yin, D., Cui, Y., Zheng, R., Li, Z., Lin, Z., ... & Chang, K. W. (2025). LLMs as Scalable, General-Purpose Simulators For Evolving Digital Agent Training. arXiv preprint arXiv:2510.14969.

---

> > > > > ### Comment · Reviewer_D1t6 · 2025-11-26
> > > > > **Response to author rebuttal**
> > > > >
> > > > > Thank you for your detailed response. I have carefully read other reviews and author response, and most of my concerns are addressed. Given that the impressive performance and the potential of environment scaling for LLM agent, I have raised my score from 6 to 8. Good luck!

---

> > > > > > ### Author Response · Authors · 2025-11-27
> > > > > >
> > > > > > Thank you for your positive feedback and support of our work! We sincerely appreciate the time and thought you put into helping us refine and improve DreamGym, so that we can share this work with the broader agent environment scaling community!

---

### Author Response · Authors · 2025-11-23

Dear Reviewers and Area Chairs,

We sincerely thank all the reviewers and area chairs for their time and thoughtful feedback on our submission. We are encouraged that the reviewers recognize the novelty and unique contribution of our work in bridging the scalability gap of training agents in real environments via reinforcement learning. We have carefully addressed each concern point-by-point in our individual responses and have made corresponding revisions in the updated manuscript. Below we summarize the key clarifications, additional experiments, and new analyses included in our rebuttal:


- **Expanded comparisons with representative RL baselines and peer approaches**
  Following the suggestions of Reviewers JoEU, PzPS, and D1t6, we have broadened our experimental comparisons:
  - We add results for representative generic RL algorithms including DAPO [1] and Dr. GRPO [2], to more clearly position DreamGym relative to SOTA RL training paradigms for LLMs. The updated results are incorporated into Table 5 in Appendix C.1.
  - We additionally compare against domain-specific RL frameworks on WebArena including WebAgent-R1 [3], WebRL [4], and DigiRL [5], which incorporate specific reward design and heavy environment-oriented engineering. We show that DreamGym matches or outperforms these frameworks while requiring no domain-specific infrastructure and far fewer real online interactions, as shown in Table 5 in Appendix C.1.
  - Following the suggestions of Reviewers D1t6 and JoEU, we also introduce a head-to-head comparison with two concurrent synthetic-training frameworks (both released later than our submission), Simia-RL [6] and UI-Simulator [7]. We report success rate, wall-clock time, and monetary cost on WebArena in Table 6 in Appendix C.2, where the results highlight DreamGym’s advantages in both performance and cost-efficiency.

- **Detailed analysis of annotation and inference costs**
  In response to Reviewers D1t6 and JoEU’s concerns about missing cost analysis, we now provide a detailed breakdown of computational and monetary costs:
  - For the annotation stage, we now report the number of trajectories, reasoning-trace annotation cost, and task-variation annotation cost for each environment in Table 7 in Appendix C.3. We also add an ablation comparing Llama-3.3-70B vs. GPT-4.1 as the annotation model in Table 8 in Appendix C.3.
  - For the agent RL data collection (inference) stage, following Reviewers D1t6 and JoEU, we report wall-clock training time, average tokens per trajectory, and per-trajectory inference time across all environments in Table 9 in Appendix C.4.
  - As suggested by Reviewer D1t6, on WebArena, we further provide a direct comparison between DreamGym and the real environment, including: average time per step, average time per trajectory, environment reset cost, total training time, and sampling failure rate in Table 10 in Appendix C.4. This analysis shows that DreamGym is substantially faster and more stable, avoids costly resets, and reduces environment hosting costs by a large margin.

- **Clarification and empirical validation of the replay-buffer update strategy**
  In response to Reviewer SnpT’s concerns regarding potential error accumulation and the replay buffer’s filtering mechanism, we have:
  - Added a detailed clarification of the replay-buffer update process, which retains only feasible transitions that lead to at least one successful continuation while discarding states associated exclusively with failure, and caps the proportion of synthetic entries through a ratio hyperparameter to prevent synthetic drift.
  - To empirically validate this design, we provide additional ablation results across different values of the synthetic data ratio on WebArena, evaluating both end-to-end success rate and state quality using GPT-4o as a judge (hallucination, consistency, diversity, informativeness). The results are reported in Table 11 in Appendix C.5, demonstrating that incorporating up to 30% synthetic data yields the highest overall agent performance while preserving state quality comparable to the purely real-data setting.

---

> ### Author Response · Authors · 2025-11-23
>
> - **More systematic and comprehensive related-work discussion**
>   In line with Reviewer JoEU’s comments on the related-work survey, we have revised Section 2:
>   - In Section 2.1 (LLM Agents Reinforcement Learning), we now systematically cover recent RL paradigms for LLM training, including DPO, PPO, GRPO, DAPO, and Dr. GRPO, and explicitly connect them to the challenges of long-horizon agent tasks (limited task sets, sparse rewards, infrastructure instability).
>   - We then discuss domain-specific RL frameworks such as WebAgent-R1 and WebRL, clarifying how they address engineering challenges (e.g., reset reliability, scalable data collection) yet remain constrained by the inherent environment challenges.
>   - In Section 2.2 (Training Agents with Synthetic Data), we broaden our discussion of synthetic expert data, model-based RL with synthetic environments, and two concurrent synthetic agent training frameworks, Simia-RL and UI-Simulator, positioning DreamGym as, to the best of our knowledge, the earliest unified, scalable framework for training LLM agents in synthetic environments that can transfer across arbitrary real-world tasks.
>
> - **Improved reproducibility: pseudocode, data filtering/cleaning, prompts, and hyperparameters**
>   To address Reviewer JoEU and PzPS’s detailed reproducibility concerns, we have:
>   - Added pseudocode in Algorithm 1 in Appendix B.2  that formalizes the full pipeline, including experience-model training, synthetic rollout generation, replay-buffer updates, curriculum-based task generation, and RL policy updates.
>   - Provided an expanded description of data filtering and cleaning for each environment in Appendix B.3, including how we collect raw trajectories, balance success/failure, and ensure diverse action coverage.
>   - Provided a complete list of prompt templates (for data filtering, annotation, experience-model inference, and policy inference) in Appendix E, which are already included in our first submission.
>   - Added comprehensive hyperparameter tables in Table 3 and Table 4 of Appendix B.1, summarizing all key configurations for reproducibility.
>   - To address Reviewer PzPS’s request for explicit task-generation examples, we also include representative cases for WebShop, ALFWorld, and WebArena in our response below.
>
>
> Thank you again for your thoughtful suggestions and efforts in helping us improve DreamGym. We hope our response adequately addresses your concerns, and we look forward to engaging in further discussions during the follow-up period!
>
> Best regards,
>
> Submission #19123 Authors
>
>
>
> [1] Yu, Q., Zhang, Z., Zhu, R., Yuan, Y., Zuo, X., Yue, Y., ... & Wang, M. (2025). Dapo: An open-source llm reinforcement learning system at scale. arXiv preprint arXiv:2503.14476.
>
> [2] Liu, Z., Chen, C., Li, W., Qi, P., Pang, T., Du, C., ... & Lin, M. (2025). Understanding r1-zero-like training: A critical perspective. arXiv preprint arXiv:2503.20783.
>
> [3] Wei, Z., Yao, W., Liu, Y., Zhang, W., Lu, Q., Qiu, L., ... & Li, L. (2025). Webagent-r1: Training web agents via end-to-end multi-turn reinforcement learning. arXiv preprint arXiv:2505.16421.
>
> [4] Qi, Z., Liu, X., Iong, I. L., Lai, H., Sun, X., Zhao, W., ... & Dong, Y. (2024). Webrl: Training llm web agents via self-evolving online curriculum reinforcement learning. arXiv preprint arXiv:2411.02337.
>
> [5] Bai, H., Zhou, Y., Pan, J., Cemri, M., Suhr, A., Levine, S., & Kumar, A. (2024). Digirl: Training in-the-wild device-control agents with autonomous reinforcement learning. Advances in Neural Information Processing Systems, 37, 12461-12495.
>
> [6] Li, Y., Inan, H. A., Yue, X., Chen, W. N., Wutschitz, L., Kulkarni, J., ... & Rajmohan, S. (2025). Simulating Environments with Reasoning Models for Agent Training. arXiv preprint arXiv:2511.01824.
>
> [7] Wang, Y., Yin, D., Cui, Y., Zheng, R., Li, Z., Lin, Z., ... & Chang, K. W. (2025). LLMs as Scalable, General-Purpose Simulators For Evolving Digital Agent Training. arXiv preprint arXiv:2510.14969.

---

### Author Response · Authors · 2025-12-03
**Letter to AC**

Dear Area Chairs,

We sincerely thank you for your time and effort in coordinating the review process for our submission. We are encouraged that the reviewers have recognized the novelty and unique contribution of our work in introducing **DreamGym**, a unified framework that bridges the scalability gap of training agents via reinforcement learning in synthetic environments. During the rebuttal, we provided extensive new experiments and clarifications to address all concerns, receiving positive feedback in return. Most notably, **Reviewer D1t6 explicitly confirmed their satisfaction with our revisions and raised their score from 6 to 8** in their final comment (*"I have carefully read other reviews and author response, and most of my concerns are addressed. Given that the impressive performance and the potential of environment scaling for LLM agent, I have raised my score from 6 to 8."*).

Overall, the reviewers found our work to be **novel, impactful, and well-executed**, particularly highlighting the framework’s ability to enable effective RL on "non-RL-ready" benchmarks like WebArena where previous methods struggle.

The major concerns raised during the review focused on **baseline comparisons**, **cost analysis**, **error accumulation**, and **reproducibility**. Our responses are summarized as follows:

**Concern 1: Expanded Baselines and Comparisons.** Reviewers (JoEU, PzPS, D1t6) requested comparisons against a broader set of RL algorithms and peer approaches.
+ **Our Response.** We significantly expanded our evaluation to include: (1) Representative generic RL algorithms (**DAPO, Dr. GRPO**), (2) Domain-specific frameworks (**WebAgent-R1, WebRL, DigiRL**), and (3) Concurrent synthetic-training frameworks (**Simia-RL, UI-Simulator**). The results demonstrate that DreamGym matches or outperforms domain-engineered systems while requiring no real online interactions, and achieves higher success rates at significantly lower costs compared to concurrent synthetic approaches.

**Concern 2: Detailed Cost and Efficiency Analysis.** Reviewers (D1t6, JoEU) requested verifiable metrics regarding annotation and inference costs.
+ **Our Response.** We provided a detailed breakdown of costs for both the annotation and RL data collection stages. We demonstrated that DreamGym incurs negligible annotation costs (<$10 per environment) and drastically reduces training time. For example, on WebArena, DreamGym completes training in **15 hours** (vs. 120 hours in the real environment) and reduces inference latency from **3.4s to 0.08s per step**, confirming its scalability and efficiency.

**Concern 3: Error Accumulation and Buffer Updates.** Reviewer (SnpT) queried the potential for error accumulation in the self-training loop and the replay buffer strategy.
+ **Our Response.** We clarified our buffer update strategy, which employs **feasibility-based filtering** (retaining only transitions leading to success) and a **controlled synthetic data ratio ($\beta$)**. We added ablation studies showing that a synthetic ratio of $\beta=0.3$ yields optimal performance without degrading state quality (verified by GPT-4o judging), confirming that our mechanism effectively prevents hallucination and error accumulation.

**Concern 4: Reproducibility and Task Generation.** Reviewers (JoEU, PzPS) requested more details on implementation and task generation.
+ **Our Response.** We added comprehensive materials to the Appendix, including **detailed pseudocode** for the full pipeline, a complete list of **prompt templates**, explicit **hyperparameter tables**, and concrete **examples of generated tasks** for WebShop, ALFWorld, and WebArena. These additions significantly lower the barrier to reproducibility.

We are encouraged by the consensus among reviewers regarding the value of DreamGym and hope this summary assists in your final assessment. Once again, we sincerely appreciate your time and effort in helping manage and improve our work!

Best regards,

Submission #19123 Authors

---

### Meta-Review · Area_Chair_Boff · 2026-01-05

**Summary:**

This paper introduces DreamGym, a framework for scalable reinforcement learning (RL) of LLM-based agents by training them in synthetic environments. The core idea is to learn a "reasoning-based experience model" that predicts state transitions and rewards in an abstract textual space, avoiding costly real-environment interactions. The framework includes an experience replay buffer and an entropy-driven curriculum task generator. Experiments on WebShop, ALFWorld, and WebArena demonstrate that DreamGym achieves competitive or superior performance to baselines while drastically reducing real interaction costs. Notably, on the challenging WebArena benchmark, it enables effective RL training where traditional methods struggle.

**Reviewer Concerns:**

Reviewer D1t6's concerns about novelty and cost were thoroughly addressed. The authors clarified that while rooted in model-based RL, DreamGym's contribution is a practical, scalable instantiation for LLM agents that prioritizes policy improvement over environment fidelity. Detailed cost analyses (annotation, inference, wall-clock time) and comparisons with concurrent synthetic frameworks (Simia-RL, UI-Simulator) demonstrated DreamGym's efficiency and superior performance, satisfying the reviewer.

Reviewer SnpT's concerns about error accumulation and buffer updates were resolved. The authors detailed the feasibility-based filtering mechanism, the controlled synthetic data ratio, and provided ablation studies showing stable state quality, effectively mitigating the risk of error propagation.

Reviewer PzPS's concerns about baselines and task generation were met. The authors added comparisons against a wide array of methods (DAPO, Dr. GRPO, WebAgent-R1, WebRL, DigiRL) and provided concrete examples of generated tasks, showcasing the framework's generality and the task generator's output.

Reviewer JoEU's concerns about missing comparisons and reproducibility were largely met. The authors expanded comparisons significantly and added substantial methodological detail (pseudocode, hyperparameter tables, prompt templates, data filtering procedures).

**Reviewer Scores:**

Reviewer D1t6: 6 -> 8. The reviewer explicitly raised their score based on the "impressive performance," addressed concerns, and the "potential of environment scaling."

Reviewer JoEU: 2 -> ?. The reviewer's concerns about comparisons, cost metrics, and reproducibility were addressed with substantial new content. While a final score change isn't recorded, the authors' rebuttal was comprehensive and directly responsive.

Reviewer SnpT: 6 -> ?. The reviewer's technical concerns about the training loop were clarified with new explanations and ablations. The response was satisfactory.

Reviewer PzPS: 6 -> ?. The requests for broader baselines and task examples were fully met. The response was satisfactory.

---

### Decision · Program_Chairs · 2026-01-26

Accept (Poster)